



# The Lattice Boltzmann Method for Wind Farm Simulations: A Review

Henry Korb[1], Jean Bastin[1], Henrik Asmuth[1], and Stefan Ivanell[1]

[1]Uppsala University, Department of Earth Sciences, Wind Energy Division

**Correspondence:** Henry Korb (henry.korb@geo.uu.se)

**Abstract.** High fidelity numerical simulations are currently a key method to gain insight into flows in very large wind farms. However, these simulations are extremely costly in terms of computational resources and the progress in computational efficiency has been outrun by the growing size of wind farms and the need for simulations. The adoption of massively parallel hardware, namely graphics processing units (GPUs), by the wind energy community has begun but the numerical structure of the Navier-Stokes equations hinders an efficient use of such hardware. That is one of the reasons, the lattice Boltzmann method has gained increasing attention in recent years. By construction, this method for simulating the Navier-Stokes equations is perfectly parallelizable and well suited for massively parallel hardware. However, as with every new method, the foundation of the method is not widely known by the wind energy community and often met with doubt. This review paper collects the various methods necessary for a potential GPU-resident wind farm flow solver based on the lattice Boltzmann method. Furthermore, it discusses various aspects of the application of the lattice Boltzmann method to wind farm flows and related flow regimes. It also identifies gaps in the current literature and aims to direct future research on the lattice Boltzmann method for wind farm flows.

## 1 Introduction

High-fidelity wind farm simulations are an essential tools for both research and industrial applications in the field of wind energy. The continuous increase in wind turbine size, combined with the expanding scale of wind farms, has led to more complex and computationally intensive simulations. In industrial contexts, it has been empirically demonstrated that the ability to complete simulations overnight is a critical benchmark for acceptance (Asmuth et al., 2023; Löhner, 2019). However, the computational demands of Large Eddy Simulations (LES) for wind energy with conventional methods are significantly too high for industrial application and limit scientific progress. To address this challenge, researchers are exploring novel approaches to conducting wind simulations.

One promising approach involves making use of the parallel processing capabilities of Graphics Processing Units (GPUs), which can significantly reduce both simulation time and associated costs. The ExaWind project (Sprague et al., 2020), featuring AMR-Wind and Nalu-Wind, attempts to port traditional approaches to GPUs. The proprietary solver GRASP by Whiffle (Schalkwijk et al., 2012) based on the Dutch Atmospheric Large Eddy Simulation is emerging as an commercial application of





LES and FastEddy (Sauer and Muñoz-Esparza, 2020) uses a numerical discretization specifically tailored to be more suitable for GPUs.

A fundamentally different approach on LES is based on the Lattice Boltzmann Method (LBM). Rooted in kinetic theory, the LBM offers extensive potential for parallelization, enabling efficient and rapid simulations with significant applicability to wind energy. Due to these advantages, it has attracted growing attention within the wind energy community. However, due
to its novelty, many aspects of it have either not been investigated or have remained unclear to large parts of the wind energy research community. Despite its advantages, it has only found limited application in wind energy to date, due to its relative novelty and its historical limitations.

The objectives of this review paper are twofold. First, we aim to collect the numerous methods necessary to implement a LES model based on the LBM for wind energy, provide a brief introduction to the underlying theory, and identify gaps in the
current literature. Second, we want to demonstrate to the research community that the LBM is a proven, accurate and efficient method for simulation of wind farm flows and familiarize the community with some of the basics of the LBM.

We limit the scope of our review to methods relevant to LES of wind farms in the atmospheric boundary layer (ABL). For any solver to be suitable for wind farm LES we consider the following criteria to be necessary:

- **Low Mach Number**: ABL flows occur at Mach numbers well below the incompressible threshold (Ma < 0.1). Therefore,
the solver does not need to simulate compressible flows.

- **Stability at High Reynolds Number** ABL flows are characterized by high Reynolds numbers ($\mathrm{Re} > 10^7$ (Stull, 1988)), thus requiring low numerical diffusivity and high stability of the numerical method of the solver.

- **Advanced Subgrid Models**: The accuracy of LES for ABL flows depends heavily on the parametrisation of subgrid-scale (SGS) turbulent fluxes (Gadde et al., 2021).

- **Wall Modeling**: Given that ABL simulations are inherently wall-bounded, turbulence scales near the surface (within the lowest 10% of the ABL (Kawai and Larsson, 2012) becomes progressively smaller. Since resolving the smallest scales near the wall is computationally prohibitive, wall models are essential for accurately predicting flow behaviour at the closest nodes from the surface.

- **Turbine Modeling**: Fully resolving rotor dynamics is computationally expensive, especially for large wind farms. Ac-
tuator models like the Actuator Line Model (ALM) (Sørensen and Shen, 2002), Actuator Disk Model (ADM) (Sørensen and Myken, 1992), and the more recent Actuator Sector Model (ASM) (Mohammadi et al., 2023) offer computationally efficient alternatives by modeling the effect of the turbine via body forces. Hence, the solver must have the capability to integrate body forces and must be stable for large force jumps over relatively small volumes.

- **Complex Terrain and Forested Areas**: Onshore wind farms are often situated in remote, forested, or hilly regions,
where terrain complexity significantly impacts wind conditions. Effective simulation requires incorporating these effects, as discussed in Elgendi et al. (2023) for complex terrains and Arnqvist (2015) for forest environments.





– **Thermal Stratification**: Atmospheric stability significantly influences critical flow parameters such as wind shear and turbulence intensity. The extensive literature on this topic (Sathe et al., 2013; van den Berg, 2008; St. Martin et al., 2016; Wharton and Lundquist, 2012) underscores the importance of incorporating thermal stratification for accurate simulations.

The remainder of this paper is structured as follows: we discuss the theory in section 2 structured into subsections: in subsection 2.1 we give an introduction to the isothermal lattice Boltzmann method and review the methods purely related to the LBM, we discuss specific aspects of modeling wind turbines in the LBM in subsection 2.2, and subsection 2.3 discusses approaches to incorporate thermal stratification; in section 3, we discuss results from various applications of the LBM and asnwer many of the most common questions facing LBM and we conclude in section 4.

## 2 The theory behind the Lattice Boltzmann Method

The following section gives an overview over the theory behind the LBM and specific aspects related to wind energy.

### 2.1 The isothermal Lattice Boltzmann Method

In contrast to conventional numerical methods used in wind energy, which solve the discretized Navier-Stokes Equations (NSE) directly, the LBM is rooted in kinetic theory. While continuum mechanics, from which the NSE are derived, describes *macroscopic* scales and individual particles are considered at the *microscopic* scale, kinetic theory addresses the intermediate, commonly referred to as the *mesoscale*. Through Chapman-Enskog analysis, the NSE can be derived from the Boltzmann equation in the limit of dense gases; the Boltzmann equation is therefore the more general equation. However, we want to stress that, although the LBM can be derived from the Boltzmann equation, the LBM solves the NSE, not the general Boltzmann equation. Historically, the LBM evolved from lattice gas cellular automata (LGCA, McNamara and Zanetti, 1988). From the LGCA, the LBM has inherited its excellent parallelizability, encapsulated by the principle: "all nonlocality is linear, all nonlinearity is local" (Geier et al., 2015a). This subsection will give a short introduction to kinetic theory, and the derivation of the LBM and describe the different models necessary and suitable for a wind farm flow solver.

#### 2.1.1 The Particle Distribution Function

The Boltzmann equation describes the evolution in time and space of the particle distribution function (PDF) $f(\boldsymbol{x}, t, \boldsymbol{\xi})$. The PDF describes the likelihood of measuring a particle with microscopic velocity $\boldsymbol{\xi} = [\xi, \upsilon, \zeta]^T$ at location $\boldsymbol{x}$ and time $t$. The ensuing paragraph follows the excellent description by Krüger et al. (2017). The moments $M$ and central moments $C$ of the PDF are defined as:

$$M_{\alpha\beta\gamma} = \int \xi^\alpha \upsilon^\beta \zeta^\gamma f \mathrm{d}^3\xi \tag{1}$$

$$C_{\alpha\beta\gamma} = \int (\xi - u)^\alpha (\upsilon - v)^\beta (\zeta - w)^\gamma f \mathrm{d}^3\xi, \tag{2}$$

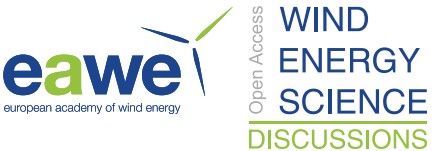

where $\boldsymbol{u} = [u, v, w]^T$ is the macroscopic velocity. Note that this notation follows Geier et al. (2015b), where the indices denote the order of the moment in that velocity component. The order of the moment is then the sum of the indices, i.e. $M_{100}$ is a first order moment, $C_{110}$ is a second order central moment and so on. The moments can be related to a variety of macroscopic quantities. The density $\rho$ and the macroscopic velocity can be recovered from the PDF by taking the zeroth and first order moments, respectively:

$$\rho = \int f \mathrm{d}^3\xi = M_{000} \tag{3}$$

$$\rho\boldsymbol{u} = \int \boldsymbol{\xi} f \mathrm{d}^3\xi = [M_{100}, M_{010}, M_{001}]^T. \tag{4}$$

The pressure $p$ has to be related to the density by an equation of state, typically the ideal gas law

$$p = \rho R T, \tag{5}$$

where $R$ is the specific gas constant of the fluid and $T$ the fluid's temperature. Second order moments also have macroscopic meaning. For example, the total energy $E$ and the inner energy $e$ can be recovered via the trace of the second order moments and central moments, respectively:

$$\rho E = \frac{1}{2}\left(M_{200} + M_{020} + M_{002}\right) = \frac{1}{2}\int |\boldsymbol{\xi}|^2 f \mathrm{d}^3\xi, \tag{6}$$

$$\rho e = \frac{1}{2}\left(C_{200} + C_{020} + C_{002}\right) = \frac{1}{2}\int |\boldsymbol{\xi} - \boldsymbol{u}|^2 f \mathrm{d}^3\xi, \tag{7}$$

And the central moments of second order and the stress tensor $\boldsymbol{\sigma}$ are related by:

$$\sigma_{x^\alpha y^\beta z^\gamma} = -C_{\alpha\beta\gamma} \ \forall \alpha + \beta + \gamma = 2. \tag{8}$$

The molecules in a gas tend towards an equilibrium state, described by the Maxwell distribution that only depends on temperature, velocity and density:

$$f^{\mathrm{eq}}(\boldsymbol{\xi}, \boldsymbol{u}, \rho) = \rho \left(\frac{1}{2\pi R T}\right)^{3/2} \exp\left(\frac{-|\boldsymbol{\xi} - \boldsymbol{u}|^2}{2RT}\right). \tag{9}$$

It can be shown that the equilibrium distribution corresponds to a fluid without viscous stresses. Therefore, the viscous stresses must be contained in the non-equilibrium part of the PDF $f^{\mathrm{neq}} = f - f^{\mathrm{eq}}$. A common way to analyze the Boltzmann equation is the Chapman-Enskog analysis. It is based on a perturbation expansion of the PDF around $f^{\mathrm{eq}}$:

$$f = f^{\mathrm{eq}} + \epsilon f^{(1)} + \epsilon^2 f^{(2)} + \dots, \tag{10}$$

with the parameter $\epsilon$ being on the order of the Knudsen number, the ratio of the length of mean free path of the particles to a macroscopic length scale.



### 2.1.2 The Lattice Boltzmann Equation

The Boltzmann equation is the transport equation of the PDF. Taking the total derivative of the PDF $f$ yields:

$$\frac{\mathrm{D}f}{\mathrm{D}t} = \frac{\partial f}{\partial t} + \frac{\partial f}{\partial x_i}\frac{\mathrm{d}x_i}{\mathrm{d}t} + \frac{\partial f}{\partial \xi_i}\frac{\mathrm{d}\xi_i}{\mathrm{d}t} \tag{11}$$

$$= \frac{\partial f}{\partial t} + \frac{\partial f}{\partial x_i}\xi_i + \frac{\partial f}{\partial \xi_i}\frac{F_i}{\rho} = \Omega, \tag{12}$$

where $\Omega$ is the collision operator. In the Boltzmann equation this collision operator has to account for all possible collisions between particles and is therefore rather cumbersome. Since the actual operator is of no interest for the LBM, it is omitted here and approximations of it will be outlined in subsubsection 2.1.3. On the left hand side, the expanded form consists of three terms. The first term is a time derivative, the second term is a convective term and the third term describes an acceleration. Accelerations and forces are related by Newtons second law, thus forces can be directly represented in the Boltzmann

equation. In comparison to the Navier-Stokes equations, no diffusion or non-linearity is found on the left hand side. All of this behaviour is contained in the collision operator. This separation of convection and non-linearity is the source of many of the advantages the LBM has over traditional CFD approaches. It can be shown via the aforementioned Chapman-Enskog analysis that the Boltzmann equation recovers the Euler equations with $f = f^{\mathrm{eq}}$ and with a first order approximation the Navier-Stokes equations, see Krüger et al. (2017, p. 26) for further details.

The lattice Boltzmann equation can be obtained from discretizing the Boltzmann equation. For details of the typical discretization on a Cartesian grid, the reader is referred to Krüger et al. (2017, p.71-98), for brevity's sake only some key results from the discretization will be discussed here. Other discretizations exist, such as for curvilinear grids, presented in Reyes Barraza and Deiterding (2020) and super-sonic speeds, for example Saadat et al. (2020), however, we have deemed them not relevant here, as we are interested in atmospheric boundary flows. Typically, the lattice Boltzmann method is understood to be

discretized on a cubic grid of nodes, with spacing $\Delta x$ and a constant time step $\Delta t$. The cubic grid is related to the fact, that the Boltzmann equation has to be discretized not only in space and time, but also velocity space. For wall-bounded flows, such as the ABL, 27 discrete velocities are recommended by Kang and Hassan (2013). This is called a D3Q27 velocity set. Another common choice is the D3Q19 set. A detailed explanation of the construction of velocity sets can be found in Shan et al. (2006). From discretizing the PDF at the discrete velocities $\boldsymbol{c}_{ijk} = [i,j,k]^T\Delta x/\Delta t$, we obtain the populations $f_{ijk}(\boldsymbol{x}) = f(\boldsymbol{x},\boldsymbol{c}_{ijk})$.

The D3Q27 velocity set consists of all possible combinations of $i,j,k \in -1,0,1$. Applying the method of characteristics to the Boltzmann equation, we can obtain the discretized Boltzmann equation:

$$f_{ijk}(\boldsymbol{x} + c_{ijk}\Delta t, t + \Delta t) = f_{ijk}(\boldsymbol{x},t) + \Delta t\Omega_{ijk}(\boldsymbol{x},t). \tag{13}$$

Again, the equation is rather simple in comparison to discretized Navier-Stokes equations. It is important to note that (13) is often split into two operations, a collision step, where the collision operator is computed and then a propagation step, where the

post-collision populations are distributed to the neighbouring nodes. This is equivalent to a split into a potentially non-linear and a non-local step. We want to emphasize, that, after a small change of variables, this approximation is already second order accurate.



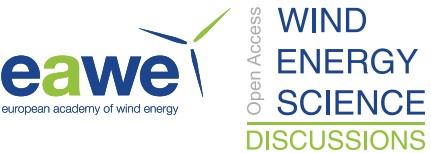

There are a few more aspects of the discretization we have to discuss. First, each velocity set leads to a speed of sound, $c_\mathrm{s}$, that is $\frac{1}{\sqrt{3}}\frac{\Delta x}{\Delta t}$ in the case of D3Q19 and D3Q27. Since the standard LBM solves the weakly compressible Navier-Stokes equations, typically restricted to flows with Mach number $\mathrm{Ma} = \frac{V_0}{c_\mathrm{s}} < 0.1$, where $V_0$ is a reference velocity, the time step is restricted to

$$\Delta t < \frac{\Delta x}{\sqrt{3}V_0}\mathrm{Ma}. \tag{14}$$

It should be noted that it is common practice in the LBM to set $\mathrm{Ma}$ to an arbitrary value, namely as high as possible while limiting compressibility effects, since it increases time step size and computational speed. Second, we have left out the force term here. Different methods of force-integration exist, which are discussed in Krüger et al. (2017, p. 231ff), but the most common approach is developed by Guo et al. (2002). The final term of (13) that will have to be discussed is the collision operator.

### 2.1.3 Collision Operators

As mentioned previously, the Boltzmann collision operator is not of interest for the LBM, rather the collision operator has to be chosen such that the LBM solves the intended equations, in our case the Navier-Stokes equations.

**The Bhatnagar-Gross-Krook Operator:** The first collision operator proposed was the Bhatnagar-Gross-Krook operator, introduced in Bhatnagar et al. (1954) to analyze the Boltzmann equation. It has the form

$$\Omega_{ijk}^{\mathrm{BGK}} = \frac{1}{\tau}\left(f_{ijk}^{\mathrm{eq}} - f_{ijk}\right). \tag{15}$$

It can be viewed as a relaxation of the populations towards the equilibrium with the relaxation time $\tau$. The relaxation time is related to the viscosity $\nu$ by

$$\nu = c_\mathrm{s}^2\left(\tau - \frac{1}{2}\Delta t\right). \tag{16}$$

While it is numerically very efficient, it is only suitable for low Reynolds and Mach numbers. With the BGK operator as a starting point, three different approaches to improve its stability and accuracy at higher Reynolds numbers have been developed. One approach is to apply different types of regularizations, while only using one relaxation time. The other approach consists of methods increasing the number of tunable parameters and transforming the distributions to moment or cumulant space. For a timeline of the development of these approaches and a theoretical and numerical comparison the reader is referred to Coreixas et al. (2019, 2020). The third branch of collision operators, the so-called entropic methods, extend the applicability of the LBM by using different velocity sets and enforcing the H-Theorem from kinetic theory, defined in Boltzmann (1872). This approach even can be extended to simulate supersonic thermohydrodynamic flows, see for example Frapolli et al. (2016). However, in the flow regime of interest here the two former methods appear much simpler and faster (Coreixas et al., 2019). We will therefore limit our review to regularized and multi-relaxation time LBMs. We also omit other ways of solving the LBM, such as the simplified and highly stable LBM by Chen et al. (2017).





**The Regularized LBM:** The possibility to improve the accuracy of the first order approximation was first shown by Latt and Chopard (2006). This method is based on the reconstruction of non-equilibrium parts of the populations through the discrete

analogue of (8). In the literature, it is often referred to as projection regularization as it can be interpreted as an projection of the non-equilibrium parts of the populations onto the Hermite basis vectors. The projection regularization approach was extended to a recursive regularization to increase stability at high Reynolds and Mach numbers in Malaspinas (2015). The recursive regularization is accomplished through a recursive relationship between the moments of the equilibrium population and the non-equilibrium parts of the PDF. An extension to include thermal and high-order LBMs was presented in Coreixas et al.

(2017). By adding a finite difference reconstruction of the non-equilibrium populations the regularized approach was extended for explicit and implicit LES in the hybrid recursive-regularized LBM by Jacob et al. (2018). It introduces a tunable parameter for hyperviscosities, which can be set to add dissipation, similar to a classical explicit subgrid-scale model. A stability analysis of the athermal models can be found in Wissocq et al. (2020). It shows the drastic increase in stability obtainable by using the regularization approaches without introducing any new parameters. One disadvantage of this method is the need to compute

additional finite differences. Thus, it relies on non-local quantities which goes against the locality paradigm of the LBM and is clearly disadvantageous in terms of computational performance.

**Multi-relaxation Times LBM:** The other large and actively developed group of collision operators originate from the idea to transform the populations to some moment-like space and apply different relaxation times for these moments. It is possible to choose different relaxation rates because only the relaxation rate of the second order moments is related to a physical property,

namely the viscosity. The first such method proposed is the multi-relaxation time method presented in d'Humières (1994). A careful analysis of the 2D case in Lallemand and Luo (2003) showed how the different relaxation rates are connected and gave some insight into the choice of the higher order relaxation rates. Following this work, relaxation in different moment-spaces were proposed, such as the cascaded LBM by Geier et al. (2006) and the central moment LBM in Dubois et al. (2015). The approach of the cascaded LBM was further developed to the cumulant LBM, introduced in Geier et al. (2015b). It exhibits

improved Galilean invariance, although the mechanism used is not limited to the cumulant LBM as the authors stress in Geier et al. (2020), is stable for very high Reynolds number flows and can be modified for fourth order accuracy in diffusion (Geier et al., 2017a, b). By adding a finite difference term even fourth order accuracy in convection can be accomplished (Geier and Pasquali, 2018). Furthermore, it is suitable for under-resolved simulations as shown in Geier et al. (2020). The method consists of transforming the populations into cumulant space. Since cumulants are statistically independent, the authors argue this is

the only moment-space where independent relaxation rates are valid. In its original form it only relies on the local quantities, however the transformation into cumulant space is cumbersome.

### 2.1.4 Boundary Conditions

One of the major difficulties in the transition from Navier-Stokes solvers to the LBM is the question of boundary conditions. In the LBM, a boundary condition describes a method of determining unknown populations at the domain boundary. Boundary

conditions can be divided into two groups, wet-node approaches and link-wise approaches. An illustration of both methods can be seen in Figure 1. Wet-node approaches assume that the boundary lies on the boundary node and the macroscopic quantities





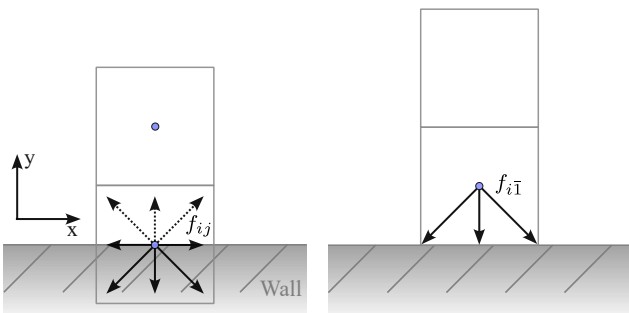

**Figure 1.** Schematic illustration of the two methods for imposing boundary conditions. The left side shows the wet-node approach, with unknown populations indicated by dashed arrows. The right side presents the link-wise approach, using Miller indices to denote the opposite y-direction.

are used to reconstruct all populations at the node. The link-wise approaches assume the boundary to lie between a solid and a fluid node. Subsequently, only the populations that would stream from the solid domain into the fluid domain have to be determined. Trying to impose velocities and density or velocity gradients necessarily leads to an under-determined system of equations as 9 populations have to be determined from 4 quantities in the case of straight walls. A plethora of closures for these systems of equations can be found in the literature.

It is easy to see that periodic boundary conditions are analogous to Navier-Stokes solvers and fairly simple to implement.

Another rather straight forward boundary condition is that of a solid wall. Several approaches exist, but the most common is the bounce-back approach (BB). As the name suggests, the populations bounce back from the wall to the node they originated from. It can be shown that this method is second order accurate if the wall is located half the cell width away from the original node. Different interpolation methods exist for walls located at arbitrary distances from the original node. By adding a source of momentum, this approach can be extended to moving walls. Further details can be found in Krüger et al. (2017, p. 175-189)

Full slip boundaries can be implemented similarly to solid walls, but instead of bouncing back, the populations are bounced forward, as described in Krüger et al. (2017, p. 206-208)

The biggest difficulty for LBM solvers lies in the open boundaries. Neither the prescription of velocities nor pressures is straightforward and this is still an active area of research. By prescribing velocity and density, it is possible to find the equilibrium populations on the boundaries. However, since the stresses at equilibrium are zero, this creates an inconsistency if the inflow is not uniform. Another way of prescribing the velocity on the boundary is by using the bounce-back method as described above and by prescribing a wall velocity normal to the wall. See Latt et al. (2008) for a detailed discussion of velocity boundary conditions and Krüger et al. (2017, p.200) for a discussion of all bounce-back boundary conditions. In classical Navier-Stokes solvers, the outlet condition typically describes a pressure, used as a reference pressure in the simulation. In the weakly compressible LBM, there is no need to prescribe a reference pressure, since the pressure is calculated from the density.





So instead a reference density is needed, which is 1 in the non-dimensionalized form. Thus, the outlet condition has to fulfil two criteria: Advection of momentum out of the domain while not exciting spurious oscillations.

The anti-bounce-back approach is similar to the bounce-back approach for solid walls, with the difference that the bounced populations have a flipped sign. To compute the bounced populations, a wall-normal velocity has to be prescribed, which has to be extrapolated from inside the domain. The approach is described in Krüger et al. (2017, p. 200). Another approach is proposed in Appendix F of Geier et al. (2015b), which relies on copying missing populations on the face of the domain from nodes inside the domain. The reflection of pressure waves which are rather common in the LBM is a common issue at open

boundaries (see, e.g., Krüger et al., 2017, p. 519). Various damping methods have been found, that filter these waves, see for example Xu and Sagaut (2013), or appendix F of Geier et al. (2015b). Another approach is the introduction of a sponge layer, a layer of increased viscosity, that dampens the pressure waves before they reach the outlet. This technique is also being used in Navier-Stokes solvers, as discussed in Colonius (2004).

### 2.1.5 Refinement

Within one domain there often exist areas with different requirements in cell size. This is usually the case near walls and, in the case of wind farm simulation, the area of the turbine and the wake. For traditional finite-difference or finite volume methods, it is no problem to change the cell size continuously, within certain bounds. This is not the case with the LBM. Since the ratio of cell width to time step determines the lattice speed of sound, changing the cell width either leads to a change in speed of sound or time step. A change in speed of sound is undesirable since it leads to refraction of acoustic waves, for example

discussed in Schönherr et al. (2011). On the other hand, changing the time step is very disadvantageous for parallelization and would require interpolations of populations. To avoid these problems, the domain can be partitioned into refinement zones. The location of the refined nodes relative to the coarse nodes differs between authors. Either some of the fine nodes coincide with the coarse nodes or all of the fine nodes are placed between coarse nodes. These approaches are called vertex-centered and cell-centered, respectively. The concept of refinement was first introduced into the LBM by Filippova and Hänel (1998a).

The authors use coinciding nodes and compute the populations for the finer grid by using a second order interpolation of populations of the coarse nodes. While this approach offers flexibility in the refinement ratio, it is not accurate or stable enough for high Reynolds flows. Based on the cascaded LBM, Geier et al. (2009) describe the so-called bubble functions, which make use of the information about derivatives of the velocity that is stored locally in the central moments. These bubble functions are second order accurate interpolations of the momentum. Thus, in theory an arbitrarily spaced and oriented fine grid could be

placed within the coarser grid. However, the authors limit their examination to a ratio of 2. The bubble functions are adopted to the cumulant LBM and extended to 3D in Kutscher et al. (2019). A different approach, that is based on the hybrid recursive regularized LBM is presented in Feng et al. (2020). Due to the enforcement of a global entropy equation, correction terms are calculated during the streaming step.



### 2.1.6 Large Eddy Simulation

The LBM can also be used for LES, which is required for the simulation of ABL flows. There exist multiple ways of accounting for the contributions of the subgrid-scale turbulence. The path towards an LES formulation specific to the LBM is described in Sagaut (2010). However, a simpler approach is to employ an eddy viscosity model and recompute the relaxation frequency with the effective viscosity,

$$\tau_{\text{eff}} = \frac{\nu + \nu_{\text{sgs}}}{c^2} + \frac{\Delta t}{2}. \tag{17}$$

Malaspinas and Sagaut prove that, in the case of the weakly compressible LBM, the filtered Navier-Stokes equations are recovered. Therefore all the models already developed for Navier-Stokes can be reused in LBM-LES. Note however, that this is only true for the weakly compressible case. For higher order discretizations the effective relaxation time is much more involved and can not easily be expressed in viscosities. A variety of SGS models have been used in the LBM, such as the classic Smagorinsky model and its refined versions, the wall adapted local eddy viscosity model (WALE) and shear improved

Smagorinsky (SISM) (Smagorinsky, 1963; Nicoud and Ducros, 1999; Lévêque et al., 2007); newer models such as the Vreman model (Vreman, 2004), the coherent structure model (Kobayashi, 2005) or QR model (Verstappen, 2011) and the anisotropic minimum dissipation (AMD) model (Rozema et al., 2015) have also been applied occasionally.

Many higher-order collision operators introduce tunable parameters, which can be used for implicit LES. For example, in Geier et al. (2020), the authors show that the cumulant LBM intrinsically accounts for the high wave number contributions.

Gehrke and Rung (2022b) introduce a model that dynamically modulates a Smagorinsky-style eddy viscosity based on third order cumulants.

### 2.1.7 Wall modelling

Due to the very high Reynolds numbers in ABLs, the flow close to the ground can not be sufficiently resolved for LES. Especially between the first node and the true location of the wall, it is necessary to model the influence of the wall. For

this purpose, a number of different relations for the velocity near the wall, called wall functions, have been found, such as Reichard's, Spalding's and Musker's, law (Reichardt, 1951; Spalding, 1961; Musker, 1979) or Monin-Obkhov Similarity Theory (Stull, 1988). When approaching the wall, eddies become smaller, thus making it difficult for turbulence models to correctly predict turbulent viscosity near the wall. Typically, turbulence models overpredict eddy viscosity near the wall. While wall modelling poses a problem in classical solvers already, those difficulties are exacerbated in LBM solvers, due to the

difficulties of prescribing shear stresses directly. Numerous wall methods have been proposed in recent years, based on a variety of approaches. In this study, we classify these methods into four distinct categories, analysing the advantages and limitations of each. We present a schematic representation of all approaches in Figure 2.

**Direct population reconstruction:** This approach relies on the direct reconstruction of the unknown population at the first wall-adjacent node. This principle was employed for the first implementation of a wall model in LBM-LES by Malaspinas and

Sagaut (2014). The authors propose to reconstruct the populations at the node closest to the wall as the sum of equilibrium and





non-equilibrium parts. The equilibrium is computed from density and velocity at the node obtained with a wall function, and non-equilibrium parts from estimating the shear stress and employing (16). One disadvantage of the model is that it requires an estimation of the turbulent viscosity near the wall for the reconstruction of the non-equilibrium parts of the populations. Since this model is based on a wet-node boundary approach, it was initially limited to straight boundaries. The approach was

extended in Wilhelm et al. (2018) to curved boundaries. First, macroscopic quantities are interpolated to an arbitrary reference point normal to the wall, then the velocity at the node is computed via the wall model and derivatives are computed from one-sided finite differences. In a final step, the equilibrium and non-equilibrium are parts of the populations are computed. Note that this model was originally developed in the context of solving RANS equations with the LBM, however, the method is equally applicable to LES. Further refinements in the computation of macroscopic quantities at the wall are presented in

Degrigny et al. (2021).

Another approach to model curved boundaries via reconstruction was proposed by Haussmann et al. (2020), where an interpolated bounce-back step is applied first, followed by a velocity correction at the boundary nodes. This correction is performed by setting the equilibrium part according to the velocity computed with the wall function while keeping the non-equilibrium parts unchanged.

**Wall Function Bounce (WFB):** WFB, introduced by Han et al. (2021a, b), is a variation of the classical bounce forward method, equivalent to a free slip boundary condition. This approach applies a drag term $F_\tau$ to the diagonally reflected populations, effectively decelerating the flow in the vicinity of the wall. This drag term is computed from the wall shear stress given by a wall function. Since the wall shear stress is imposed directly, no turbulent viscosity is needed. However, it is also unclear whether the model can be extended to a D3Q27 lattice or curved boundaries.

**Immersed Virtual Wall (IVW):** Kuwata and Suga (2021) introduces yet another approach for wall modelling, named the specular reflection bounce-back method. It is very similar to the WFB approach, however the drag force is applied to all populations, not just those coming from the wall. Because the specular reflection is difficult to extend to curved boundaries, the authors propose a virtual wall layer approach, where an arbitrary number of virtual fluid layers is introduced below the wall. A drag force $F_{\mathrm{wm}}$ to replace the viscous force $F_\nu$ with the force due to the wall shear stress $F_\tau$ is imposed at the node

neighbouring the wall:

$$F_{\mathrm{wm}} = F_\tau - F_\nu. \tag{18}$$

A bounce back scheme is applied at the lowest level of virtual fluid nodes. Thus the model can easily be applied at curved boundaries and any lattice and does not require the computation of a turbulent viscosity. However, additional nodes need to be included in the domain. Furthermore, by applying the force to all populations the shear near the wall is not correctly calculated.

**Wall velocity approach:** The final class of approaches considered relies on determining a fictitious wall velocity, $\boldsymbol{u}_{\mathrm{w}}$, which is then used to modify the BB scheme. Pasquali et al. (2020) leverage the additional information about the stress tensor provided by the cumulants of the populations to compute a skin friction coefficient and from that $\boldsymbol{u}_{\mathrm{w}}$. This approach is local to the first node off the wall, and is applicable to curved boundaries. However, it does rely on a relation of skin friction coefficient and bulk velocity. As noted in the study, it should be applied exclusively with the cumulant collision operator, as less accurate methods





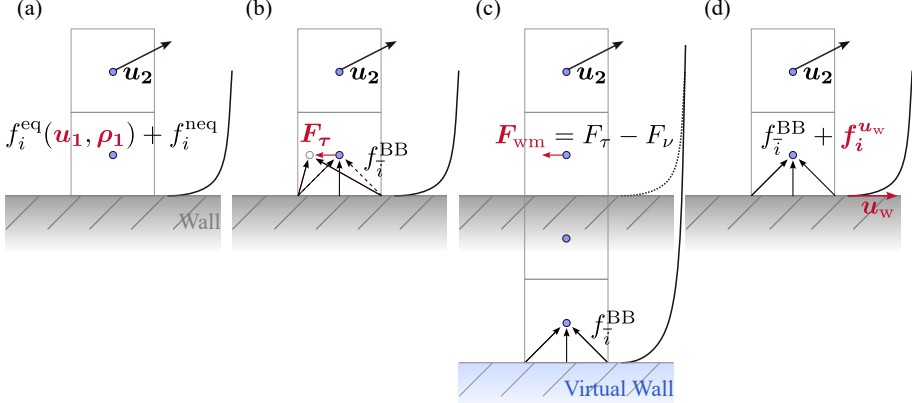

**Figure 2.** Schematic representation of the different wall modeling approaches in LES-LBM. Symbols in red indicate the required outputs for model implementation, while black denotes known information. (a) Population reconstruction approach, (b) wall function bounce, (c) immersed virtual wall method, (d) wall velocity approaches.

fail to provide the necessary precision for an accurate stress tensor approximation. A viscosity is needed to relate cumulants and stresses.

The approach introduced by Asmuth et al. (2021), called the inverse momentum exchange method (iMEM), computes a wall velocity so that the momentum exchanged between the wall and the node closest to the wall matches the wall shear stress computed from a wall function. This method is independent of the collision model and no turbulent viscosity is required. However,
Asmuth et al. only present a formulation for straight boundaries, an extension to curved boundaries is not yet available, but the authors note that it is possible.

### 2.1.8 Complex Geometries

The main difficulty of applying the LBM to complex boundaries is the Cartesian grid, on which most LBMs are based on. A variety of strategies exist to either circumvent or address this issue. A simple method is a stepwise approximation of the geom-
etry and the use of simple bounce-back methods. This introduces an error in the approximation of the geometry, while keeping the boundary condition nominally second order accurate. By refining the region around the boundary, the approximation error can be reduced.

If a better approximation of the geometry is necessary, interpolated bounce-back methods (IBB) are another option. IBB are based on the fact, that populations travel exactly $\Delta x$ during one time step. Therefore a fictitious population is created by
interpolation, that will travel $\Delta x$ and end up at the boundary node. Methods differ on the interpolation scheme. The original IBB proposed by Bouzidi et al. (2001) uses a linear or quadratic interpolation. The resulting scheme is independent of the collision operator and stable, however, it is not local if the boundary is closer than $\frac{\Delta x}{2}$. It is also not mass-conserving, as shown in Krüger et al. (2017, p. 447). Furthermore, the distance at which the fictitious population has to be interpolated has to be known. In the case of a stationary boundary this only has to be done once, but if the boundary moves, this computation has to be repeated





every time step. An immersed solid wall approach is presented in Feng et al. (2019b), that computes the equilibrium and non-equilibrium populations based on interpolation of macroscopic quantities. It requires multiple interpolations of macroscopic quantities and is therefore likely to be computationally expensive. However, an extension to include wall models appears straight forward, although not explicitly mentioned in the study.

Another class of boundary conditions is based on extrapolation methods. There exists a variety of methods with different advantages and disadvantages. The interested reader is pointed to Krüger et al. (2017, p. 455-463).

A different approach to implementing complex geometries is via the immersed boundary method (IBM). The idea of IBM, introduced in Peskin (2002), is to model the geometry via Lagrangian marker points that exert a force onto the fluid. The geometry is therefore completely decoupled from the lattice. This makes it suitable for complex and moving geometries, for example in particle-laden flows. Since the IBM models the boundary via forces, the same methods can be used for Navier-Stokes solvers and LBM solvers.

IBMs can be split into two groups: explicit methods and direct forcing. In explicit methods the marker points exert a force proportional to the virtual deformation of the boundary due to the fluid. This introduces a proportionality constant, which is effectively a spring constant, that has to be determined. High values of the spring constant lead to higher accuracy of the geometry but can introduce instability, while the opposite is true for low values, see Krüger et al. (2017, p. 474f). The direct forcing approaches construct a force such that the velocity post-collision has the correct value. A variety of ways to compute this force exist. The original variant by Feng and Michaelides (2005) computes the force explicitly by computing the force of the fluid on the node and then imposing a balancing force. Another approach, called the implicit velocity correction-based IB-LBM, computes the force of all marker points at the same time and is introduced in Wu and Shu (2009). This requires the inversion of a matrix of size $N^2$, where $N$ is the number of marker points. However, this matrix only depends on the geometry of the boundary. Thus, if the boundary is rigid, it only has to be done once. The previously mentioned approaches all aim at imposing a no-slip condition at the boundary. However, if IBM is to be used to model a complex terrain in high-Reynolds flows, a wall-model has to be included as well. We could not find an example of IB-LBM used with wall model in the literature. The problem has been addressed with classical Navier-Stokes solvers, though, for example in Chester et al. (2007).

## 2.2 Simulating wind turbines and farms in the LBM

To simulate wind turbines and farms in a flow solver, the effect of the turbine on the flow must be accounted for. Full rotor simulations that resolve the geometry of the rotor require either the mesh to move with the rotor or to remesh when the rotor has moved. Both approaches have been applied in the LBM and we will discuss these examples in subsection 3.7.

However, often the exact flow at the blades is not of interest and the rotor and tower can be modelled via body forces using actuator line (Sørensen and Shen, 2002) or actuator disc methods (Sørensen and Myken, 1992). Here we only want to highlight aspects of these methods in relation to the LBM. More information on these methods can be found in, for example Mikkelsen (2003); Troldborg (2009); Martínez-Tossas et al. (2015). Since both methods rely solely on imposing forces on the flow, they can easily be introduced to the LBM.





Generally, the actuator line is considered more accurate. However, it is usually also more costly because it limits the maximal size of the time step, since the tip of the actuator line should not move more than one cell per time step (Troldborg, 2009). With $\lambda = \frac{\omega R}{V_0}$ as the tip-speed ratio and $\omega$ as the rotational speed of the turbine, the time step is limited by

$$\Delta t < \frac{\Delta x}{V_0 \lambda}. \tag{19}$$

With $\lambda \approx 5 - 10$ for modern turbines, this limit is significantly lower than the CFL condition, namely $\Delta t < \frac{\Delta x}{V_0}$, which is a typical stability-criterion in implicit incompressible Navier-Stokes solvers. Since the LBM is an explicit method, the time step size is significantly smaller, as described in subsubsection 2.1.2. Thus, for $\lambda < \frac{\sqrt{3}}{\mathrm{Ma}} \approx 17$, the time step limit for ALM is more relaxed than the limit for the LBM itself, assuming $\mathrm{Ma} = 0.1$. This makes the application of the ALM very attractive for the LBM. Indeed, to our best knowledge, only the ALM has been used in combination with the LBM so far, as we discuss further in subsection 3.7.

To summarize, wind turbines can be simulated in the same way in the LBM as in Navier-Stokes-based solvers. However, one of the main drawbacks of the ALM in implicit Navier Stokes solvers is not present in the LBM.

## 2.3 Thermal Stratification

Since the LBM is based on kinetic theory and, as was shown before, the total energy $E$ and internal energy $e$ can be computed from the distribution functions, it would be fair to assume that the LBM can be extended to include temperature as well. While the full Navier-Stokes-Fourier equations can be derived from the Boltzmann equation, a discretization of such equations would require the use of more than 27 discrete velocities in three dimensions and the interaction between nodes that are not direct neighbours, as shown, for example in Qian (1993). These so-called multi-speed methods suffer from high computational cost and instability (McNamara et al., 1995). However, in the ABL, it is sufficient to solve the Navier Stokes equations and an advection diffusion equation (ADE) for temperature and couple them via the Boussinesq approximation (Stull, 1988). A variety of approaches have been developed to solve such equations with the LBM. The so-called hybrid methods discretize the advection-diffusion equation directly with finite difference or finite volume methods, while the double distribution function (DDF) approach solves the ADE using a second set of populations and an altered LBM scheme. A recent review of both approaches can be found in Sharma et al. (2020), therefore we include here only studies with particular relevance to our application or more recent studies.

The derivation of the hybrid methods is rather straight forward and was first introduced in Filippova and Hänel (1998b) and improved again in Filippova and Hänel (2000). Further progress is made in Lallemand and Luo (2003) by utilizing an MRT collision operator and a thorough analysis of instabilities in the thermal LBM. Three-dimensional simulations for convective flows were first conducted in Mezrhab et al. (2004) and showed good results for low Rayleigh and Reynolds numbers. An implementation of the hybrid model for a multi-GPU was proposed in Obrecht et al. (2012) and tested in Obrecht et al. (2013). Obrecht et al. utilize an Euler-forward time integration and a finite difference scheme based on all direct neighbours, analogous to the discretization of the LBM. The authors stress, based on Lallemand and Luo (2003), that the finite difference operator needs to have the same symmetry as the LBM stencil operator to improve stability. However, no further explanation is





provided. A model combining the hybrid recursive regularized LBM with a finite volume solver for scalars is presented in Feng et al. (2019a, 2021). The finite volume solver employs a Euler-forward discretization for the time integration and the MUSCL scheme (Van Leer, 1977) and central differences for the advective and diffusive terms respectively. The authors use this model to simulate two active scalars, namely humidity and potential temperature.

In the DDF approach, the ADE is solved via the LBM. The temperature, $\theta$ or any other scalar, is the zeroth moment of the populations $g_{ijk}$

$$\theta = \sum g_{ijk}. \tag{20}$$

First order moments of equilibrium are related to the advective flux

$$\theta \boldsymbol{u} = \sum \boldsymbol{c}_{ijk} g_{ijk}^{\mathrm{eq}}. \tag{21}$$

Note that, in contrast to the LBM for the NSE, the first order moments of the populations and the first order moments of the equilibria are not equal. A similar relation of relaxation time $\tau$ to diffusivity $\alpha$ as (16) exists:

$$\alpha = c_{\mathrm{s}}^2 \left( \tau - \frac{1}{2} \Delta t \right). \tag{22}$$

The application of the LBM to solve other than the Navier-Stokes equations and specifically ADE began shortly after the LBM was first developed. One early exploration of the LBM to solve ADE can be found in Wolf-Gladrow (1995). The author shows
that an LBM scheme very similar to the one used for fluid flow is able to approximate the ADE. For a more extensive explanation of the LBM for ADE the reader is referred to Krüger et al. (2017, p.297-321), which also covers boundary conditions. Different collision operators can be used and many of the operators proposed for the momentum equations have been adopted for ADE. An overview of proposed collision operators can be found in Gruszczyński and Łaniewski-Wołłk (2022), highlighting that only few models have been proposed for three dimensions. Furthermore, the overview misses the model proposed in Yang
et al. (2016), which is based on the factorized cascaded LBM. The same model is used by Adekanye et al. (2022) to simulate an active scalar.

In a wall-modelled LES, the temperature model also needs to employ a SGS and a wall model. A variety of models that compute an eddy diffusivity exist, see for example Stoll et al. (2020); Gadde et al. (2021). Making use of the effective diffusivity approach, analogous to the effective viscosity approach, and (22) enables the use of such models in the DDF approach. In finite-
difference solvers, the application of a wall model is straight forward by applying Monin-Obukhov theory to compute the heat flux for a given stability, as done in Feng et al. (2019a, 2021). In contrast to the Navier-Stokes model, it is possible to prescribe a heat flux with a bounce-back approach the same way as a wall velocity is specified. Another approach for a wall model is proposed by Kuwata and Suga (2021). The authors propose to use the same immersed virtual wall approach as described in subsubsection 2.1.7 to model the wall heat flux as a source term on the first node.

In summary, the LBM can and has been augmented by a separate solver for temperature, either based on finite differences or an altered LBM scheme. Nevertheless, not many models have been proposed, especially in three dimensions. The development of the LBM for thermal simulations lags behind that of the isothermal LBM, especially for wall modeled LES.





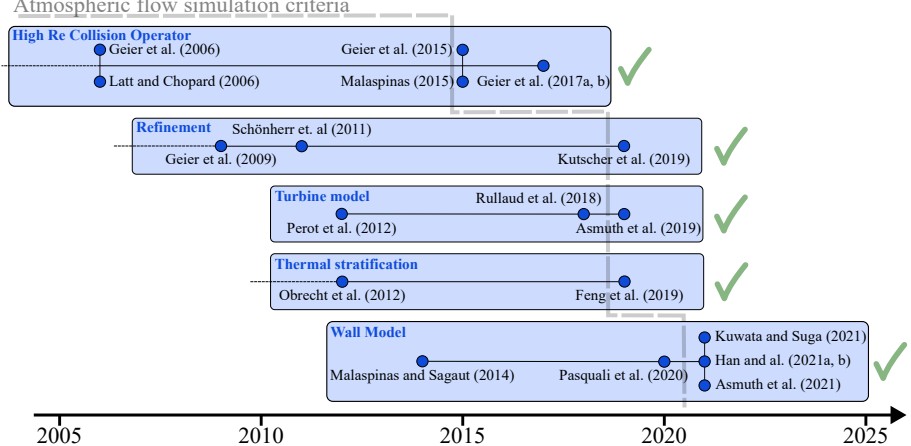

**Figure 3.** Overview of recent key advancements in research on simulating atmospheric flows using LBM-LES, presented as a timeline

## 2.4 Summary of the available methods in the lattice Boltzmann method

The application of the LBM in wind energy is a relatively recent research field, with many aspects still actively being explored.

To a large extent, this is due to the fact that a number of crucial advancements in the fundamentals of the LBM for wind energy applications only happened during the past decade. An overview of these milestones is collected in a timeline shown in Figure 3. Today, the LBM is equipped with several accurate and sufficiently robust collision operators to handle high Reynolds number flows as needed for atmospheric boundary simulations or wind turbine wake flows. The same subgrid models that are used in Navier Stokes based solvers can be used in the LBM, thus enabling LES of atmospheric boundary layer flows. A number of

wall modeling approaches and refinement methods exist to enable large computational domains while reducing computational cost. Wind turbines can be simulated either as full rotor simulations or using actuator line models, therefore also making it possible to simulate wind farms with a large number of rotors. Methods for extending solvers to include thermal stratification, forest canopies and complex terrain exist, enabling the simulation of realistic conditions wind turbines are exposed to. Thus, the methods necessary for the LBM to become a comprehensive, reliable and fast LES tool capable of meeting the increasing

simulation requirements of wind energy research and industry, exist.

## 3 Discussion

In the following section, we discuss different applications of the LBM in relation to wind farm flows. We gather studies to demonstrate how various aspects of the LBM have been examined, beginning from its capability to simulate highly resolved turbulent flows to atmospheric boundary layers and wind farm flows. At the same time we identify areas where more research

is needed. We structure our discussion around a set of questions we have asked ourselves or have been asked during the last years as we have been conducting research on the LBM.





### 3.1 Is the LBM suitable for turbulent flow simulations?

All flow phenomena relevant for wind farm simulations, be it ABL flows or wind turbine wakes, are highly turbulent. Thus, a numerical scheme used for these applications needs to be suitable for very high Reynolds numbers. Interestingly, this ca-
pability is still frequently questioned, particularly from people less familiar with the method, including the wind energy LES community. (It should be noted that we can only refer to anecdotal evidence for this scepticism. Examples are discussions about these capabilities at conferences, or requests from reviewers to add fundamental validation cases to application-oriented papers even though various previous publications have covered these topics extensively.) Indeed, the initial BGK collision model, and even later MRT approaches do suffer from numerical instabilities at high Reynolds numbers. However, these issues have
been remedied by modern collision models, as outlined in subsubsection 2.1.3. In the following we discuss these capabilities and provide numerical evidence to support that claim. We do so by highlighting several studies that perform direct numerical simulations (DNS) and slightly under-resolved simulations of turbulent wall-bounded flows.

Validations of LBM solvers against benchmark DNS of various flow conditions have been successfully carried out since the early 1990s, affirming the LBM's suitability for bulk flow studies.

Jahanshaloo et al. (2013) provide a comprehensive review of studies validating LBM using fully or pseudo-spectral DNS. The pioneering work by Benzi and Succi (1990) examined turbulent flow using the BGK operator within a two-dimensional square domain with periodic boundary conditions, evaluating LBM's efficiency relative to spectral methods. At low Reynolds numbers, the study found good agreement in both enstrophy and energy time series, as well as in the energy spectra. The authors also noted that, while both methods had comparable computational costs, LBM demonstrated improved scaling at
higher grid resolutions.

Lammers et al. (2006) conducted resolved simulations of fully developed channel flows at a friction-based Reynolds number of $\mathrm{Re}_\tau = \frac{u_\tau H}{\nu} = 180$, where $u_\tau = \sqrt{\frac{\tau_\mathrm{w}}{\rho}}$ is the friction velocity based on the wall shear stress $\tau_\mathrm{w}$ and $H$ is the channel half-height. They used the BGK method on a D3Q19 lattice, comparing the results with those from pseudo-spectral solvers. The results showed remarkable agreement, concluding that LBM is as reliable as Chebyshev pseudo-spectral codes for DNS of
turbulent flows, with the authors stating, "*This removes any doubt that LBMs have inferior performance in resolved DNS.*" Furthermore, they emphasized the LBM's numerical efficiency, achieving a 5x speed-up over the pseudo-spectral solver.

A comparison of various collision operators (BGK with D3Q19 and D3Q27, MRT, and cascaded LBM) was conducted by Freitas et al. (2011). The study found that the BGK model showed better agreement with reference results in simulations of turbulent channel flows at $\mathrm{Re}_\tau = 200$. However, this finding contrasted with results from their lid-driven cavity test, where the
BGK models exhibited instabilities at higher Reynolds numbers, while MRT and cascaded models maintained good agreement. They concluded that universal stability assessments across methods remain challenging. This highlights that the MRT operator was not able to improve the stability of the LBM in general.

Another comparison of collision operators was performed in Nathen et al. (2018), this time of the BGK, MRT and regularized (Latt and Chopard, 2006) operators. In studying dissipation of the Taylor-Green vortex at varying Reynolds numbers and
resolutions, they find that the BGK operator is accurate if the simulation is sufficiently resolved, while the MRT model is



accurate in under-resolved simulations, yet becomes unstable at too high resolutions. The regularized model is always stable but very dissipative.

The work of Gehrke et al. (2017) compared the performance of BGK, MRT, and cumulant collision operators in DNS of turbulent channel flows at moderate Reynolds numbers ($\mathrm{Re}_\tau = 180$). This study found that DNS simulations with all three collision operators produced excellent results, and that the cumulant operator was stable even at coarser grid resolutions, where BGK and MRT models exhibited non-physical oscillations. Gehrke et al. emphasized the resolution-dependent damping nature of the cumulant model, noting that it acts similarly to an inherent subgrid-scale model.

In conclusion, LBM has been used to simulate turbulence for 35 years and has been shown to be as accurate as traditional approaches based on discretizing the NSE, while being numerically significantly more efficient. However, it is only with the recent introduction of more advanced collision operators, especially the regularized and cumulant collision operators, that stability and accuracy at lower resolutions have become feasible in LBM.

### 3.2 Is the LBM applicable to LES simulations? Which lattice and collision operators are most suitable? How are subgrid-scales accounted for?

A large number of LES studies has been conducted with LBM, utilizing explicit or implicit methodologies to model the subgrid-scales and employing various lattices and collision operators. In implicit LES, it is assumed that the effects of subgrid scale motions are modeled accurately enough by the collision operator without additional explicit models. Again, we only highlight a few studies of particular importance to wind farm LES.

A review of early studies of LES-LBM can be found in Jahanshaloo et al. (2013). The authors gather a large collection of studies with different collision operators and subgrid-scale models. They find that, in general, boundary treatment is of high importance for accurate results.

In Kang and Hassan (2013), the authors examined the influence of the velocity discretization on the results of simulations of square ducts and round pipes. They found that a D3Q19 lattice is sufficient if the axes of the lattice align with the main axes of the walls. However, the lack of isotropy in the lattice negatively impacts the results if the duct is rotated 45° or in the round pipes. Similar problems related to the isotropy of the lattice have been reported in Asmuth et al. (2020b) when simulating wind turbine wakes.

The MRT D3Q19 model was applied together with the Smagorinsky SGS model to simulate a channel flow at $\mathrm{Re}_\tau = 180$ by Wu et al. (2011). They furthermore simulate a passive scalar identified as temperature with a MRT D3Q7 model. The results match the reference DNS data reasonably well considering the differences in the numerical approaches.

Wang et al. (2014) also simulate a channel at $\mathrm{Re}_\tau = 180$, using a BGK D3Q19 model with the Smagorinsky model for LES and DNS. They find an excellent agreement with the reference data.

Gehrke and Rung (2022a) present results from simulations of a periodic hill with the cumulant LBM at different resolutions. They show that the well-conditioned parametrised cumulant LBM is capable of performing implicit LES for highly-resolved LES. However, at very coarse resolutions the damping of higher order cumulants does not suffice and warrants the use of an additional SGS model. In a subsequent study (Gehrke and Rung, 2022b) the same authors developed a cumulant-based SGS





model along a resolution sensitive regularization. The authors simulated turbulent channel flows at $\mathrm{Re}_\tau = 180, 550, 2\,000$ with 12, 24 and 48 nodes per channel height $H$ using the cumulant LBM with a Smagorinsky SGS model and a wall function fitted to data from DNS. The authors find excellent agreement in mean velocities and shear stress even at low resolutions of 12 nodes per half height, with the exception of the highest Reynolds number, where the mean velocities in the bulk are consistently underpredicted in the low resolution case. Turbulence energy production is accurately predicted throughout the channel with

the exception of the first or first two nodes. An additional simulation at $\mathrm{Re}_\tau = 5\,200$ also agrees excellently with reference data obtained from DNS.

    Spinelli et al. (2023) compare different collision operators and turbulence models. The MRT and HRR models are tested together with the Vreman and WALE SGS models and the cumulant model is applied without any explicit SGS model. They simulate the flow past a cylinder at $\mathrm{Re} = 3\,900$, based on the cylinder diameter. The cumulant operator consistently outperforms

other collision operators significantly and matches reference data very well. The authors report that the MRT model is unstable with a D3Q27 lattice and they therefore employ the MRT model with a D3Q19 lattice.

    Overall we find that LBM-LES has been applied successfully to wall bounded flows and even the most simplistic collision operators can produce accurate results at high resolutions. However, more advanced collision operators, especially the cumulant LBM can produce highly accurate results also for more complex cases. If walls are aligned with the lattice directions, good

agreement can be found with D3Q19, however, if the walls are misaligned or strong rotational features are present, D3Q27 is generally more suitable. At high resolutions, the implicit damping from the cumulant and HRR method suffice to achieve good results, however, at coarse resolution, explicit SGS models are required. However, in a direct comparison the cumulant collision operator outperforms all other collision operators in terms of accuracy.

### 3.3   How do the different wall models for LBM perform in LES simulations?

Even with a suitable collision operator and SGS model, wall models are required to reduce the computational cost of LES, be it of full rotor simulations or the ABL. As discussed in subsubsection 2.1.7, a number of models have been proposed in the literature but we have not yet discussed how they perform.

    The wall model approach introduced in Malaspinas and Sagaut (2014) is tested in the same paper in simulations of turbulent channel flows. The flow is modeled using a D3Q19 MRT LBM with Smagorinsky SGS model and the Musker's law is applied.

Simulations at $\mathrm{Re}_\tau = 950, 2\,000$ show very good agreement in mean velocity with reference data. At $\mathrm{Re}_\tau = 20\,000$ the agreement is not as good but still satisfactory given the used resolution. Feng et al. (2021) validates the generalized reconstruction method presented in Wilhelm et al. (2018) for simulations of atmospheric boundary layers using `ProLB` and the HRR model by simulating a neutral, pressure-driven boundary layer. The mean velocities compare well to the log law and reference data in the vicinity of the wall. However, the momentum fluxes are consistently underpredicted compared to a range of reference

cases. We will discuss further results presented in that publication later on.

    Haussmann et al. (2019) compared different velocity boundary conditions and wall functions using a D3Q19 BGK model with a Smagorinsky turbulence model and a similar wall modelling approach as Malaspinas and Sagaut (2014) but employ a three-layer wall function. They find that the reconstruction of the populations based on Guo's extrapolation scheme is signifi-





cantly more accurate than the simple equilibrium reconstruction in simulations of turbulent channel flows at $\mathrm{Re}_\tau = 1\,000$. At
$\mathrm{Re}_\tau = 2\,000$ and $\mathrm{Re}_\tau = 5\,200$, results deteriorate at low resolutions and at least 20 nodes per channel half height are required.
A systematic examination of wall function, SGS model and collision operator is conducted by Spinelli et al. (2024), using the
wall model approach from Haussmann et al. (2019). A turbulent channel flow at $\mathrm{Re}_\tau = 1\,000$ is simulated with three different
resolutions, namely 10, 20 and 40 nodes per $H$. The wall functions used comprise the Musker's, Reichardt's laws and the
Power-law. We only want to report the findings on the collision operator here, where again the cumulant LBM without an
explicit SGS model consistently outperforms the HRR and MRT models also tested.

The partial slip velocity based model by Pasquali et al. (2020) is validated in the same publication with simulations of
turbulent channel flows at $\mathrm{Re}_\tau = 950, 2\,000, 16\,000$ and with resolutions of 10 and 20 nodes per $H$. They employ the cumulant
collision operator without an explicit SGS model. They compare both approaches for computing the wall shear stress and
find that computing the wall shear stress from Musker's law results in good agreement to reference data for $\mathrm{Re}_\tau = 950$ and
$\mathrm{Re}_\tau = 2\,000$ while the cumulant based wall model approach yields quite large deviations. At the highest Reynolds number
($16\,000$), the cumulant based approach yields better agreement at higher resolutions.

In Kuwata and Suga (2021), the authors compare both wall model approaches proposed in the same article to DNS of a
turbulent channel flow at $\mathrm{Re}_\tau = 5\,200$. They employ a D3Q27 MRT model, the Smagorinsky SGS model and Musker's law.
Both approaches yield good results for first and second order statistics, however the immersed virtual wall approach consistently
under-predicts the mean velocity. Due to the limitations of the specular reflection approach to straight walls, only the immersed
virtual wall approach is further investigated. At low resolutions of 10 nodes per channel half height, the accuracy of mean
velocity prediction is reduced, while the shear stress is still in very good agreement to the reference solution. The model
also performs well at lower ($\mathrm{Re}_\tau = 500$) Reynolds numbers, whereas at higher ($\mathrm{Re}_\tau = 10\,000$) the under-prediction of mean
velocity increases. Finally, the effect of the thickness of the virtual wall is assessed and it is observed that a lower thickness
yields better agreement with the reference data. Xue et al. (2023) also implement the specular reflection method by Kuwata
and Suga (2021) and apply a D3Q19 MRT together with the Smagorinsky SGS model to simulate turbulent channel flows at
$\mathrm{Re}_\tau = 1\,000, 2\,000, 5\,200$ with 10, 20 and 30 nodes per $H$ (note that in their publication, Xue et al. denote the channel half
height $\delta$ and the channel height $H$). The wall shear stress is computed from Reichardt's law. Notably, they employ a synthetic
turbulence generation method at the inflow. Far enough downstream of the inlet the results agree well with DNS reference data
in mean velocities and stresses, even at low resolutions.

The WFB approach is evaluated in Han et al. (2021b) by simulating a turbulent channel flow at $\mathrm{Re}_\tau = 640$ and $\mathrm{Re}_\tau = 2003$
with a D3Q19 MRT and the Smagorinsky SGS model. The wall shear stress is computed from Spalding's law. Results show an
improvement over not using a wall model, however, there is a consistent overprediction of mean velocity at the lower Reynolds
number. At the higher Reynolds number the error reduces. The wall model is further validated in Han et al. (2020) against
measurements of the flow passed a rectangular block with a Reynolds number based on the block width of $40\,8000$. SGS
stresses are computed with the WALE model. The comparison to reference simulations and experimental data shows good
agreement in mean velocities and TKE, however the authors report numerical oscillations attributed to the collision operator.
In Han et al. (2021a), the effect of the wall model is examined in the same setup as the previous studies. They find that the





application of the wall model reduces the grid requirements and is able to produce results of the same quality as a simple
bounce back scheme at significantly coarser resolutions, reducing computational cost.

Finally, Asmuth et al. (2021) validates the wall model approach introduced in the same study in an isothermal pressure-driven boundary layer. The cumulant LBM in conjunction with the AMD SGS model and Monin-Obukhov Similarity Theory are used and the flow statistics are compared to results from a pseudo-spectral solver. Different methods to determine the wall shear stress are compared in terms of their effect on the log-layer mismatch. The authors show that the elevated Schumann-
Götzbach model of Maronga et al. (2020) yields the best results. In a subsequent grid sensitivity study, excellent agreement of mean quantities and stresses is found and even higher order statistics show decent agreement. The iMEM approach was further examined in Gehrke and Rung (2022b), which we already partially discussed in subsection 3.2. Recall, that the authors simulated turbulent channel flows at a range of Reynolds numbers and resolutions and found excellent agreement in the bulk. Only at the highest Reynolds number did they find consistently lower velocities than the reference. Furthermore, the mean
velocity and turbulence energy production at the first node off the wall is consistently too low for the two higher Reynolds numbers.

To conclude this section, we find that many wall model approaches have shown good agreement in the simple case of a turbulent channel flow. However, a consistent and direct comparison of the different wall modelling approaches has not yet been conducted. Additionally, the effect on wall models from wall function, collision operators, and subgrid scale models has
not been explored in detail either. Furthermore, most wall models were only ever applied at Reynolds numbers significantly lower than what is found in the atmospheric boundary layer, with the exception of iMEM model and the reconstruction method. Finally, not all wall models can be extended to complex geometries and comparisons of wall models in complex geometries are missing entirely.

### 3.4 How are complex geometries handled in LBM in the context of turbulent flows?

Dealing with complex geometries, such as complex terrain surrounding wind farms, poses a number of challenges. Since the LBM is based on a Cartesian grid, it does not have the same flexibility to adapt the grid to the terrain as a finite volume method. However, as we discussed in subsubsection 2.1.4, interpolated boundary conditions can be used that approximate the shape of a geometry more accurately than a simple stair case approximation. Another challenge is the use of wall models. Some of the approaches presented in subsubsection 2.1.7 can inherently be used to model to interpolated boundaries, others can be adapted,
but that is not possible for all approaches. Finally, the data layout of a solver has to be suitable for efficient representation of complex geometries.

An early example of complex geometries represented in a GPU-resident solver can be found in Bernaschi et al. (2010). The solver employs the indirect addressing scheme that can also be found in modern solvers such as `VirtualFluids`, allowing for an efficient representation of arbitrary geometries in a fashion suitable for GPUs. Jin et al. (2015) uses a highly resolved,
D3Q19 BGK-based simulation to study the effect of roughness on channel flows. The authors highlight the efficiency of the method, consequently allowing an even higher resolution. A more recent publication considering the flow over a hill can be found in Schubiger et al. (2020). The authors assess the ability of LBM to predict the flow over a well studied benchmark





using the open-source, CPU-resident solver `Palabos`, utilizing only interpolated bounce-back boundaries without a wall model. The authors find reasonable agreement between the LBM, reference simulations conducted with RANS and DES and the experiments, despite the simplistic modelling setup. A comparison of the computation time shows that the LBM solver is about 5 times faster than the DES solver. The study highlights the continued need for an accepted wall modelling approach for complex geometries. As already mention in subsection 3.2, Gehrke and Rung (2022a) simulate the flow over a periodic hill using cumulant LBM with interpolated bounce-back boundaries but without turbulence or wall model at different resolutions. At the highest resolution the results match the reference solutions very well and different Mach numbers also show to have little influence on the quality of the solution. When using lower resolutions the quality of the solution deteriorates.

The extended reconstruction method for curved boundaries is used in conjunction with the HRR model to simulate flow around a NACA0012 airfoil in Degrigny et al. (2021), and excellent agreement to reference data is observed. In addition to the turbulent channel flow, Haussmann et al. (2019) also conduct simulations of a Coriolis mass flow meter using the wall model, bounce back and interpolated bounce back. Note, that unless the interpolated bounce back is applied, all walls are approximated with a staircase approximation. Nevertheless, the wall model yields generally good agreement with the measured pressure drop across the mass flow meter.

In Kuwata and Suga (2021), the authors apply the IVW model described in subsubsection 2.1.7 to the periodic hill test case in addition to the channel flow discussed in subsection 3.3. Musker's law is used to compute the shear stress, and a D3Q27 MRT combined with the shear improved Smagorinsky model is used for the bulk flow. The results of the simulations of the periodic hill mostly agree well with the reference solutions, however, the recirculation areas exhibit significant differences in the mean velocities and Reynolds stresses. Overall, the results are of similar quality as those reported by Gehrke and Rung (2022a) with the same resolution, who did not use a wall model.

Overall, we find that representation of complex geometries is well established, however, simulation of real-world complex terrain remains sparse. Furthermore, only two studies use a wall model in combination with complex geometries. This clearly presents a gap in the current research and is likely related to the fact that many wall modeling approaches have only been formulated for straight boundaries.

## 3.5 Is the LBM suitable to simulate large domains and the atmospheric boundary layer?

Despite the lack of wall models in complex geometries, a rapidly growing number of studies of wind in the atmospheric boundary layer have been presented in recent years. Many of these studies concentrate on flows in urban areas, which reduces the need for wall models and therefore circumvents the current limitations. However, these studies demonstrate the suitability of the LBM for wind energy research and are therefore also included here.

The earliest application of LBM to urban flows was reported in Fan et al. (2004), which is also the first implementation of LBM on a cluster of GPUs. The BGK operator without any turbulence model is used. The methodology is very simplistic, yet many more recent studies use essentially the same methodology, albeit with the inclusion of a turbulence model. The authors focus on the computational performance and report almost 50 MNUPS on a cluster of 32 NVidia GeForce FX 5900 Ultra



GPUs, each with 128 MB of memory. We mention this study to highlight how far the methodology and hardware have come in the last 20 years It also demonstrates the early embrace of GPUs by the LBM community.

Modern applications of LBM to urban flows began with Onodera et al. (2013), where simulations of a $10\,\text{km} \times 10\,\text{km}$ domain with a grid spacing of $1\,\text{m}$ of the urban area of Tokyo are presented. A D3Q19 BGK model is used, walls are modelled with the
bounce-back scheme and the coherent structure SGS model is used. Scaling tests up to $1\,000$ GPUs are performed and good scalability is obtained. Additionally, a mesh with $10\,080 \times 10\,240 \times 512 = 5.28 \times 10^{10}$ nodes is simulated on $4\,032$ GPUs. It marks one of the largest LES simulations still to this day. The same setup is used by Ahmad et al. (2017) and Inagaki et al. (2017), presenting results on wind gust index and turbulence statistics and structures respectively. Further improvements of the model with respect to scaling across a very large number of GPUs are presented in Onodera et al. (2018) and Onodera
and Idomura (2018), using adaptive mesh refinement and techniques to reduce internode communication. These studies utilize the cumulant collision operator. They report excellent computational performance and decent scalability across a very large number of GPUs, highlighting the LBM's suitability for very large simulations.

Watanabe et al. (2020) discusses simulations of plant canopies based on central moment collision operators. The forest canopy is represented with a drag force model. Comparisons with reference simulations confirm that the approach is capable
of reproducing the canopy flow very accurately.

The study by King et al. (2017) employs a D3Q19 BGK model to simulate the flow through a building facade. The comparison with experiments and a traditional CFD code is within expected variation, while the LBM approach significantly reduces the computation time.

In Lenz et al. (2019) the cumulant LBM solver `VirtualFluids` is used to study the feasibility of real-time simulations
of urban flows by comparing to measurements made in the Basel UrBan Boundary Layer Experiment. Boundaries are represented with the bounce back method and mesh refinement is used to reduce the computational cost. Good agreement with the measurements is reported and quality criteria for this simulation are reached.

In contrast to the previous studies, Jacob and Sagaut (2018) utilize the HRR collision model implemented in `ProLB` to simulate an urban flow. No subgrid scale model is used, the study mentions the use of a wall model but does not give any
details regarding its implementation. The results are compared to field and wind tunnel measurements in several points and show a fair agreement. Further studies of urban and micrometeorological flows are presented in Jacob et al. (2021).

Buffa et al. (2021) also uses the HRR approach with the wall model from Malaspinas and Sagaut (2014) and a synthetic eddy model for the inlet to study the wind loads on a single high-rise building. The authors find very good agreement with experimental data, given an adequate setup.

A study examining forest canopy modelling in the LBM was conducted in Shao et al. (2022). The study employs a BGK D3Q19 model with a Smagorinsky model. The forest is modelled with a drag term. A simulation of a boundary layer with a forest on the ground are compared to reference simulation and reasonable agreement is found in the mean velocities, while noticeable discrepancies in the fluctuations are present.

Overall, we find a large increase in the number of studies over recent years. Most studies employ the D3Q19 MRT model
with a Smagorinsky subgrid-scale model. However, groups with a longer history tend to more sophisticated collision operators





and the D3Q27 models. The HRR model is also well-established but only implemented in a single solver. Furthermore, despite claims from various authors, the actual computational efficiency of HRR is not reported in any of the papers reviewed here. Different approaches to thermal coupling are used throughout the literature, with DDF and hybrid approaches being used about equally and neither show great advantages nor disadvantage over the other. Most studies reviewed do not employ a wall model, and only two wall models have been validated in simulations of an atmospheric boundary layer. Furthermore, none of the reviewed studies utilize a precursor method and instead rely on uniform velocity or periodic boundary conditions. Since the resolution and domain size are very similar to wind farm simulations, LBM is shown to be suitable for simulations of wind farms.

### 3.6 How do the two thermal LBM approaches perform and has the LBM been applied to stratified atmospheric boundary layers?

An isothermal description of the atmospheric boundary layer is often sufficiently accurate for simulating individual wind turbines or small farms, however, the role of thermal stratification on various aspects of wind farm flows is gaining much attention. Therefore, a potential LBM solver should also have the capability to simulate thermally stratified flows. As discussed in subsection 2.3, two approaches are commonly used to simulate thermally stratified flow, the DDF and the hybrid approach. As the review by Sharma et al. (2020) notes, the vast majority of thermal LBM applications are limited to two-dimensional flows.

The study by Wu et al. (2011) simulated the transport passive scalar in a turbulent channel flow using a DDF approach with the MRT collision operator as mentioned previously. They compare their results with DNS simulations and show a reasonable agreement.

A similar methodology is used by Ren et al. (2018), however, the scalar is coupled via the Boussinesq approximation and the Vreman SGS model is used. Simulations of a turbulent channel flow with at $\mathrm{Re}_\tau = 180$ and varying Rayleigh numbers are compared with reference data obtained from DNS and the authors find good agreement, even in second order statistics.

A more advanced collision model for the second populations, namely a cascaded model on a D3Q15 lattice, is employed in Hajabdollahi and Premnath (2018) to study natural convection at Rayleigh numbers up to $10^5$ by means of DNS. They find very good agreement with reference data. Guo et al. (2023) use a DDF approach to conduct a DNS of a stably stratified flow. They employ the BGK collision operator for both distributions, with a D3Q7 lattice for the temperature and a D3Q19 for the momentum and couple the temperature via the Boussinesq approximation. The results agree very well with DNS results obtained with a pseudo-spectral solver. A comparison of the computational performance reveals that the pseudo spectral solver is significantly more efficient, due to looser requirements in mesh generation and larger time-steps. Kuwata and Suga (2021) present the only simulation of a DDF approach that also includes a wall model. However, the scalar is passive. The results of a turbulent channel flow agree well with the reference data.

Watanabe et al. (2021) simulates the transport of a passive scalar through a plant canopy with the DDF approach by augmenting the model presented in Watanabe et al. (2020) with a second distribution function modelled with a single relaxation time operator. Simulation results are compared to that of a Navier-Stokes based solver and excellent agreement is found in the



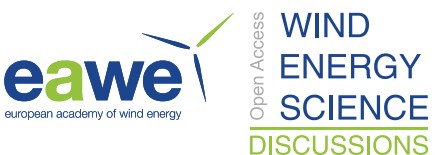

concentration of the scalar. An extension of the model used in Wang et al. (2020a) to simulating stratified flows over a ridge is
presented in Wang et al. (2020b). The temperature field is simulated using a D3Q7 MRT model. Results exhibit the expected
physical behaviour and compared well to laboratory and numerical results. However, the shown flow fields exhibit significant
oscillations.

In Onodera et al. (2021), the cumulant method method is augmented with a finite different solver for temperature and
concentration. The advection diffusion equations are discretized using second order finite differences in space and explicit
Euler forward in time. The evolution of a plume of concentration in an urban environment is simulated. The simulation is
driven by large scale meteorological date obtained from the Weather Research and forecasting model. Results are compared to
results from a measurement campaign and decent agreement is found.

In Feng et al. (2019a), `ProLB` is extended with a hybrid approach to solve temperature and humidity equations with a finite
volume approach as described in subsection 2.3. Excellent agreement to reference data is reported. The previously mentioned
Feng et al. (2021) gives an extensive description of the models used in `ProLB` and presents validation studies. Under stable
and convective conditions results agree well with the reference data despite lower resolution. Various other benchmark cases
with more complex conditions are reported as well with fair agreement. Jacob et al. (2021) applies the method described in
Feng et al. (2021) to a range of benchmark cases of urban and micro-meteorological flows and report decent to good agreement
throughout. The authors also report high computational efficiency, however, metrics are not provided.

To summarize, we find that the double distribution function and hybrid approach have been used to model temperature fields
and have been applied to model stratified atmospheric boundary layers. While subsection 2.3 showed that there is much more
theoretical discussion of the DDF model, the hybrid approach is preferred in application to stratified boundary layer flows.

### 3.7   How has the LBM been applied to LES of wind turbines and wind farms?

So far, we have discussed many individual aspects of LES related to wind energy. However, combining all of the individual
methods and models, such as a turbine model, wall model, and SGS model in a single solver is not always straight forward.
Furthermore, some models depend, for example, on specific collision operators and are maybe incompatible with others.
Nevertheless, we found a number of publications employ the LBM to study either single wind turbines or wind farm flows.
Interestingly, wind turbines were first simulated as full rotor simulations and actuator models were only introduced later on.

The first application of LBM to wind turbine simulations was conducted in Perot et al. (2012). The authors simulated a fully
resolved NREL Phase VI model with the commercial LBM solver `powerFLOW` and compared the results to experimental data
and found good agreement of blade forces and wake quantities. Xu (2016) conducted simulations of up to three NREL Phase
VI model turbines using the same solver. The results were compared to experimental data and qualitatively good agreement was
found, however, a quantitative comparison was not reported. Deiterding and Wood presented results obtained from conducting
rotor-resolved simulations of the NREL Phase VI model turbine and a Vestas V27 in Deiterding and Wood (2016a) and
Deiterding and Wood (2016b), therefore reporting the first simulation of an, albeit small, full-scale turbine. They employ the
`AMROC` solver, which includes adaptive mesh refinement and the a third-order scheme for the equilibrium distribution. They
report good agreement to the experimental data in all cases.



The first simulation of a wind turbine using an actuator line method was presented in Rullaud et al. (2018) using a 2D
LBM solver based on the MRT collision operator and validating against measurements of two vertical axis wind turbines. A
first validation study of a 3D LBM simulation coupled to the ALM was presented in Asmuth et al. (2019), employing the
cumulant LBM implemented in the GPU-resident LBM solver `elbe` without an explicit SGS model. The NREL 5MW turbine
in a uniform inflow is studied. Parameter studies of the smearing width and the Mach number are presented and compared to
results obtained with the Navier-Stokes solver `EllipSys3D`. The results of the velocity deficit in the wake show excellent
agreement. Asmuth et al. (2020b) uses the same methodology albeit with a Smagorinsky SGS model with uniform and turbulent
inflow, providing a more thorough examination of the wake and again excellent agreement to the reference solution obtained
with `EllipSys3D` was found. Compressibility in LBM-ALM simulations was studied in Asmuth et al. (2020a). The effect
on the wake was found to be negligible, although changes in density were observed in the vicinity of the actuator line. The
methodology was compared to measurements and other solvers in Asmuth et al. (2022). Two benchmark cases are defined based
on measurements from the DanAero experiments, featuring two 2.3 MW turbines. The LBM simulations feature the iMEM
wall model. The authors conclude that the models were able to capture integral quantities such as torque and thrust well,
even in waked inflow and highlight the difficulty of generating inflow conditions for LES based on met-mast measurements.
The LBM model performs similar to the other solvers presented in the benchmark. The same methodology, but implemented
in the GPU-resident LBM solver `VirtualFluids` is further validated in Korb et al. (2023), where we present validation
against the SWiFT benchmark, including turbine response and wake measurements. The setup utilizes a precursor to generate
realistic inflow, the induction correction model proposed by Meyer Forsting et al. (2019) and iMEM wall model. We find very
good agreement in the turbine response and good agreement with the wake measurements, on par with the results obtained by
traditional methods in the original benchmark by Doubrawa et al. (2020). In Asmuth et al. (2023), we examine the requirements
for industrial adoption of LES in terms of capabilities and computational performance and compare the results with a simulation
of 64 DTU 10 MW turbines (see Bak et al. (2012) for the definition) using the same methodology as described in Korb et al.
(2023).

Schottenhamml et al. (2022) present an implementation of the ALM in the LBM code `waLBerla` and compare it to simula-
tions conducted with `SOWFA`. The study also employs the cumulant LBM in a combination with the Smagorinsky model. The
actuator line model is modified to use two alternative formulations for the Gaussian smearing kernel. Simulations of the DTU
10 MW reference turbine are compared. Excellent agreement of blade forces is reported, while the velocity deficit in the wake
differs slightly. The study features scaling experiments to up to 40 GPUs and finds that `waLBerla` scales well to multiple
GPUs across a number of nodes, provided the workload is large enough. The approach is further developed in Schottenhamml
et al. (2024), where the algorithm is detailed and validation via comparisons with the NewMexico experiment is presented.
Furthermore, the excellent scaling of `waLBerla` to a total of 120 GPUs is demonstrated, thus proving is suitability to simulate
even very large wind farms, although the scaling tests only feature a single turbine.

A further implementation of an ALM in a cumulant LBM solver is presented in Watanabe and Hu (2024). The authors
present results of a weak scaling test, scaling from 4 to 256 GPUs, assigning each GPU the same number of fluid nodes and
a single turbine. They demonstrate excellent scaling of the fluid solver. However, the calculations related to the ALM are not





parallelized and therefore do not scale well. Furthermore, a simulation of a single Vestas V80 Turbine with a diameter of 80m is presented. The results compare fairly well to results from a finite-difference Navier-Stokes solver. Finally, results from simulating 8 Vestas V80 turbines in a row are presented and the importance of a rotor-speed controller is demonstrated. Most recently, an actuator line embedded in a rotating mesh was presented by Ribeiro and Muscari (2023). Simulations of the NREL Phase VI rotor and the NREL 5MW turbine with `powerFLOW` are presented and achieving excellent agreement with reference simulations. However, the presented case is somewhat unusual in that the domain includes a very fine mesh around the blade, with resolutions much higher than is typical for ALM simulations. The authors also conclude, that the approach is feasible for highly resolved simulations but might not merit its application in the typical scenario of a coarse mesh. The same sliding mesh approach was used in Ribeiro et al. (2025), comparing blade resolved simulations and highly resolved actuator line simulations to measurements obtained from a rotor in a water channel. The study finds excellent agreement between the blade resolved simulations and the actuator lines. Furthermore, asymmetric rotors are examined and the effect of asymmetry on tip vortex breakdown.

The high computational efficiency of LBM makes it a prime candidate for use in combination with data driven and machine learning methods. Two studies have utilized the LBM in combination with machine learning approaches. In Korb et al. (2021), we used the methodology developed in Asmuth et al. (2020a) to create a training environment for a deep reinforcement learning approach to optimize a wind farm controller. While we ultimately failed to improve the wind farm performance, we showcased how LBM enables new, data-driven methods. In a second study, Asmuth and Korb (2022), a similar setup was used to generate training data for a convolutional neural network to predict the average velocity and turbulence intensity in the wake of a single turbine. We found excellent agreement with the test data, while the computational time to infer a single case was negligible.

To conclude, a variety of different approaches for using the LBM for wind farms have been presented in the literature, with the most thoroughly studied approach being the cumulant LBM in combination with an actuator line. However, many of these studies are conducted in idealized conditions. Recently, studies in more realistic conditions have been presented, including wall models, and realistic inflow conditions, either through synthetic turbulence or precursor-successor setups. Furthermore, two studies simulating a large number of turbines have been presented and proven the feasibility of LBM for simulating large wind farms. Even under more realistic conditions, the method remained computationally very efficient and is thus a viable candidate for industrial application of LES. However, there was no study simulating thermally stratified atmospheric boundary layers and wind turbines.

### 3.8 Which computational performance can be expected from an LBM solver and are they really faster than Navier-Stokes based solvers?

As highlighted multiple times throughout this paper, the exceptional parallelization capabilities of the LBM makes it particularly well-suited for GPU implementation. Since the first application of LBM on a GPU cluster in 2004, substantial advancements in hardware have led to a significant increase in computational power. Million node/lattice updates per second (MNUPS/MLUPS) are a common measure of computational efficiency in the LBM community. MNUPS can also be converted to



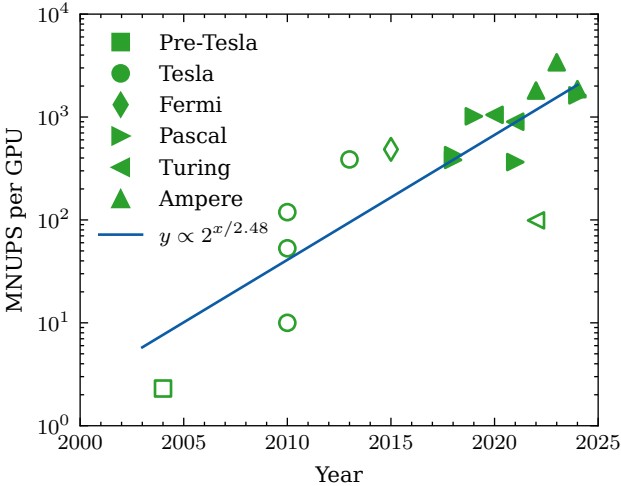

**Figure 4.** Evolution of computational performance per single GPU over time, distinguished by GPU architecture generation. Unfilled markers indicate the use of a D3Q19 stencil, while filled markers represent the use of D3Q27.

calculate the ratio of simulated time to wall time ($T_{sim}/T_{wall}$) by a simple relation:

$$\frac{T_{sim}}{T_{wall}} = \frac{\Delta t P}{N} = \frac{\Delta x \mathrm{Ma} P}{\sqrt{3} V_0 N}, \tag{23}$$

where $\Delta t$ is still the size of the time step, $P$ is the computational performance and $N$ is the number of grid points and the
second expression is obtained by employing (14). Figure 4 illustrates the evolution of computational performance, expressed in MNUPS per GPU, over time and with a distinction within GPU architecture generation. The data displayed is gathered from a compilation of LBM studies mainly focused on atmospheric flow simulations on GPUs, and which we have already discussed or presented earlier in this paper (see Table A1 for a list of the included studies). We clearly observe a steady increase in computational performance and that the current state of the art performance is 2000 - 3000 MNUPS. Therefore, given (23)
and assuming $\Delta x = 10$m, $V_0 = 10$m/s, $N = 50 \times 10^6$, $P = 2 \times 10^9$MNUPS and an artificial Ma $= 0.1$, $\frac{T_{sim}}{T_{wall}} = 2.3$, thus simulating more than twice as fast as realtime, which is in line with values reported, for example, in Asmuth et al. (2023). Furthermore, we find that the D3Q27 lattice has now replaced D3Q19 as the preferred choice. We have also computed a best fit logarithmic regression, indicating that on average computational performance has doubled approximately every 2.5 years. While Figure 4 depicts the computational performance per GPU, it is also important to address the efficiency of multi-GPU
configurations. In early studies (Fan et al., 2004; Onodera et al., 2013), multi-GPU implementations suffered from significant performance losses. However, advancements in parallelisation algorithms have substantially mitigated these inefficiencies, rendering the associated losses almost negligible in modern implementations (Onodera et al., 2018; Schottenhamml et al., 2022). Thus LBM is especially suited for very large simulations. Furthermore, there has been a very rapid increase in the available memory per GPU, thus enabling fairly large simulations to be conducted on a single node, as noted, for example in
Asmuth et al. (2023), where a single data center GPU was used to simulate a wind farm consisting of 64 turbines.





Direct comparisons to Navier-Stokes-based solvers in terms of computational performance remain sparse to this day. Furthermore, it is difficult to establish how such a comparison should be carried out since a traditional finite volume code has very different requirements than an LBM solver. The same quality of solution typically requires more cells in an LBM solver due to the fact that all cells have a fixed aspect ratio and stretching is not possible. This difficulty is exacerbated by the fact that many

LBM solvers run on GPUs, requiring few compute nodes, whereas most Navier-Stokes solvers run on massively parallel CPU systems.

In terms of time to solution, Asmuth et al. (2020b) compares to results from `EllipSys3D`, a CPU based finite volume solver and finds that the wall time of the LBM simulation run on a single consumer GPU is only a third of the FV solver, executed on an compute cluster. In terms of processor time, a reduction on the order of $10^3$ is found. Schottenhamml et al.

(2022) compare a CPU and GPU solver to SOWFA and report reductions in wall time of factor 73 and 473, respectively. However, wall time is strongly dependent on the amount of processors used. A more robust measure is the energy to solution, which is total electric energy required for the solver to arrive at the solution. In Korb et al. (2023), we compare to many different solvers in terms of energy to solution and find that our solver reduces energy to solution by 2 to 3 orders of magnitude. Yet, it still requires an order of magnitude more energy than a RANS solver.

So far, every comparison has shown that the LBM on GPU is significantly more efficient than a CPU based Navier-Stokes solver. Comparisons to GPU based Navier-Stokes solvers have not yet been performed and are difficult to do based solely values reported in literature. A fair comparison would have to be done on the same or at least very similar hardware, and simulating a clearly defined benchmark case. Furthermore, the exact metric on which to evaluate the benchmark will have to be established, since the LBM has other limitations on the spatial and temporal discretization as finite volume or finite difference

solvers. Therefore, solvers must be compared in terms of metrics taking into account the trade-off between computational effort and accuracy, such as the energy to solution vs. accuracy metric presented in Doubrawa et al. (2020).

## 4 Conclusions

This review has gathered the methods necessary to implement a solver for wind farm flows in the atmospheric boundary layer based on the lattice Boltzmann method. We have described various collision operators for the bulk flow, necessary boundary

conditions, subgrid-scale and wall modelling approaches and how wind turbines can be represented in the flow. We presented a short overview over methods to extend the isothermal LBM to thermally stratified flows. Finally, we gathered applications of the previously described methods, highlighting relevant studies of wall bounded flows, employing both DNS and LES, complex geometries, thermally stratified flows and reviewed simulations of single wind turbines, wind farms and atmospheric boundary layer flows.

We find that a variety of methods suitable for high Reynolds number flows exist and that the LBM has been successfully applied to such flows many times. Subgrid-scale models from the Navier-Stokes world can be readily applied in the LBM, and the LBM is well suited for simulating turbulence, with both DNS and LES. However, wall-modeling posed a challenge for a long time, but a variety of approaches have been proposed in recent years and successfully tested, even in isothermal





atmospheric boundary layers. While complex geometries are relatively simple to simulate with the LBM, only two approaches
to incorporate wall models have been extended to curved boundaries and applications of these methods in the context of
atmospheric boundary layers remain sparse. The actuator line method is well suited to the LBM due to the small time step
of the LBM and has been used in various studies of single wind turbines and wind farms of up to 80 turbines. However the
actuator disc has not yet been applied in the LBM. Approaches to simulate thermal stratification exist but we could find only
a small number of studies actually using these approaches in three-dimensional flows. One exception is a number of studies
conducted with the solver *ProLB*, which has been used to simulate thermally stratified atmospheric boundary layers in a number
of studies. The numerical performance of the LBM has steadily improved with improved hardware and the LBM outperforms
Navier-Stokes-based solvers consistently due to its parallel nature.

In summary, we find that all of the methods required for wind farm simulations with LBM-LES have been developed
and validated and that LBM-LES is well suited for large scale simulations, especially using massively parallel hardware.
Nevertheless, no simulation combining all of the discussed features, that is a wind farm in a thermally stratified atmospheric
boundary layer, has been conducted as of yet.

*Author contributions.* HK – Conceptualization, Investigation, Data curation, Writing – Original Draft; JB – Investigation, Writing – Original
Draft, Visualization; HA – Writing – Review & Editing; SI – Writing – Review & Editing, Supervision

*Competing interests.* The author reports no competing interests.

**Appendix A: Computational Performance**





**Table A1.** Publications listing achieved computational performance on NVidia GPUs.

| Year | Reference | Lattice | GPU | MNUPS/GPU[*] |
|---|---|---|---|---|
| 2004 | Fan et al. | D3Q19 | GeForce FX | 2.3 |
| 2010 | Bernaschi et al. | D3Q19 | GeForce 8600 | 10 |
| | | | Tesla C870 | 53 |
| | | | 2x Tesla S1070 (8x GT200) | 119 |
| 2013 | Obrecht et al. | D3Q19 | Tesla C1060 | 387 |
| 2015 | Obrecht et al. | D3Q19 | 7x Tesla C2075 | 485 |
| 2018 | Onodera et al. | D3Q27 | 196x Tesla P100 | 426 |
| 2018 | Onodera and Idomura | D3Q27 | Tesla P100 | 383 |
| 2019 | Lenz et al. | D3Q27 | Tesla P100 | 1016[**] |
| 2020 | Asmuth et al. | D3Q27 | GeForce RTX 2080 Ti | 1050 |
| 2021 | Asmuth et al. | D3Q27 | GeForce RTX 2080 Ti | 900 |
| 2021 | Onodera et al. | D3Q27 | Tesla P100 | 365 |
| 2022 | Schottenhamml et al. | D3Q27 | 8x A100 | 1809[**] |
| 2022 | Shao et al. | D3Q19 | GeForce RTX 2070 | 99 |
| 2023 | Asmuth et al. | D3Q27 | A100 | 3404 |
| 2024 | Watanabe and Hu | D3Q27 | Tesla P100 | 1624 |
| 2024 | Schottenhamml et al. | D3Q27 | A100 | 1866 |

[*]When available, the performance of a single GPU in single precision was reported. However, some studies only provided the performance of GPU clusters. In such cases, we estimated single-GPU performance by dividing the cluster's performance by the number of GPUs, assuming no parallelization losses. Although such losses have been significantly reduced in recent years, they are not entirely negligible, making some of the reported figures conservative estimates.

[**] Some inconsistencies have been observed in the reported performance of their simulations. However, the performance values remain within a credible range.

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
