# Peer review of "The Lattice Boltzmann Method for Wind Farm Simulations: A Review"

_Wind Energy Science, 2025_

## Referee Comment (RC2)

**Comments on the following paper: "The Lattice Boltzmann Method for Wind Farm Simulations: A Review"**

October 29, 2025

**1 General comments**

This paper provides a comprehensive overview of the current state of the Lattice Boltzmann Method (LBM) within the wind energy community. It is overall well-written and clearly structured, and will likely serve as a valuable resource for interested researchers and practitioners. In my opinion, the work is suitable for publication after minor revisions.

**2 Discussion by themes**

- **Performance:** The discussion on performance could be expanded slightly. For instance, the following two references, although focused on CPU implementations, could provide useful context:
  - https://doi.org/10.1177/10943420211006169
  - https://doi.org/10.1016/j.compfluid.2021.104946

If available, including performance estimates of a code such as FastEddy (https://doi.org/10.1029/2020MS002100) in Figure 4, or at least discussing them in the text, would be valuable. Finally, the issue of memory boundedness and the need for advanced streaming patterns to reduce memory requirements could also be discussed in this section.

- LBM for ABLs: The review effectively highlights the potential of LBM for simulating kilometrescale domains. However, I would suggest adding brief discussions on the following aspects:
  - Meso-scale simulations: LBM is likely constrained to micro-scale simulations due to its reliance on isotropic meshes, making meso-scale applications inefficient. This point seems to be missing from Section 3.5.
  - Coupling with meso-scale models: Related to the previous item, it would be useful to discuss
    the potential need for coupling LBM with meso-scale models and possible approaches for
    achieving this.
  - The study https://doi.org/10.1063/5.0039516 introduces an anelastic approximation within an LBM framework. This work, and the potential advantages of such an approach, merit at least a brief mention and discussion.
  - A genuine question: are there intrinsic limitations within the LBM framework that hinder simulations of non-neutral atmospheres, or the inclusion of humidity and other relevant processes? These might relate to the constant sound-speed assumption, the weakly compressible formulation, or the idealized thermodynamics typically adopted.

**• LBM theory:**

- Another question: higher-order stencils (beyond order 27) are not discussed. Could they offer any advantages for atmospheric flow simulations?
- Many Navier-Stokes-based ABL solvers include transport equations for humidity (or other scalars) and for  $k_{SGS}$ . Implementing these in LBM, especially alongside a double-distribution-function (DDF) approach, might be prohibitive due to memory constraints. This trade-off could be worth mentioning.

**• Turbulence models in LBM:**

- The focus on LES is well justified for atmospheric flows, but a short discussion on the potential use of RANS-type eddy-viscosity closures in LBM, and their possible efficiency benefits, would strengthen the review.
- In Section 2.1.6, it would be helpful to comment on how the choice of subgrid-scale model affects performance. Not all SGS models are fully local, and some rely on additional transport equations for  $k_{SGS}$ . In a DDF framework, this would require solving yet another set of populations, significantly increasing memory usage—this limitation deserves mention (see also the related note in the LBM theory subsection).
- In Section 3.2, a dedicated discussion on the use of fourth-order limiters in the cumulant space could be beneficial. It is often argued that these limiters behave as implicit SGS models and therefore should not be used in combination with explicit SGS closures.

**3 Small comments**

- L.82: Avoid using superlatives such as "excellent description."
- L.94: It would be interesting to discuss the feasibility of incorporating more complex equations of state within LBM.
- L.175: Replace "an" with "a."
- L.260: The statement that LES is required for ABL flows may be too strong; RANS models could also be employed.
- L.282: Referring to "turbulence models" here creates some ambiguity. Since the discussion focuses on LES, "subgrid-scale models" would be more appropriate.
- Section 2.1.6: Is there a recommended approach overall?
- Section 2.2: I may be misunderstanding this part, but in my view, the much smaller time steps in LBM are primarily due to the sound-speed-related restriction, rather than to the choice of explicit versus implicit schemes. This point could be clarified.
- L.659: Could you elaborate on the assumption that urban flows reduce the need for wall models? Boundary layers developing along buildings are likely to be severely under-resolved.
- L.739: Typo "different" should replace "difference."
- L.742: Typo "date" should be "data."
- A recent paper addressing wind farms in non-neutral ABLs could be included in the discussion: https://papers.srn.com/sol3/papers.cfm?abstract\_id=4441495.

---

## Author Comment (AC1)

The paper gives a review on the usage of the lattice Boltzmann method in wind farm simulation. Instead of presenting data, the various aspects of wind farm simulation, large scale LES, boundary layer functions, actuator line-models, stratified atmospheric boundary layers are discussed in textual form. Actual data is only shown for performance with different GPUs.

I belief that the paper fulfills the standards of this forum and can be published with minor revisions, albeit more quantitative discussion based on detailed data would certainly be preferable. The discussions also appear to be skewed towards a specific lattice Boltzmann model and it is not entirely clear whether this is due to the popularity of the cumulant model in the wind engineering field in general or the popularity of that models with the authors in particular.

We have given particular attention to the cumulant model since we perceive it as the most popular model. To provide a more quantitative discussion of the different collision models, we have added the new figure 4, showing the number of publications featuring each collision operator from a systematic literature review.

Some specific points listed below require some attention:

l. 82: "paragraph follows the excellent description by Kruger et al.", not saying Kruger is not excellent but scientific publications profit from avoiding judgements of this kind.

We have adjusted the manuscript accordingly.

Line 141 mention Strang splitting and/or integration along characteristics to explain 2nd order convergence.

We now mention both methods.

l. 144 "First, each velocity set leads to a speed of sound" how does the velocity set lead to a speed of sound?

We have clarified the statement.

l. 210 depending on the geometry of the boundary, the system of equations might either be over- or under-specified.

We have clarified the statement.

l.219 bounce forward only works for straight boundaries

We have clarified the statement.

l. 239 sponge layers do not primarily dampen waves. They eliminate eddies that would CAUSE waves when they interact with pressure boundary conditions. Non-reflective BCs are not an alternative to sponge layers or vice versa as both act on different problems.

We have clarified the statement.

2.2 Simulating wind turbines and farms in the LBM $\rightarrow$ confusing $\rightarrow$ Simulating wind turbines and wind farms in the LBM

We have clarified the statement.

l. 660: why does the simulation of urban flows reduce the need for wall models?

We have clarified the statement. Urban flows are typically dominated by drag forces of bluff bodies instead of aerodynamically rough walls.

l.739 method method

We have removed the duplicate.

l. 893: actuator disk has not yet been applied: An actuator disk was used in the Ph.D. thesis of Schoenherr for a ship propeller.

We have weakened our statement. However, we don't believe the method used by Schönherr should be considered an actuator disk. From personal communication and reviewing the original source code used we know that Schönherr applies a method akin to, for example, the wall model approach from Malaspinas et al. (2014). Hence the propeller does not act as a forcing term but rather sets the velocity.

---

## Author Comment (AC2)

Comments on the following paper: "The Lattice Boltzmann Method for Wind Farm Simulations: A Review" October 29, 2025

**1 General comments**

This paper provides a comprehensive overview of the current state of the Lattice Boltzmann Method (LBM) within the wind energy community. It is overall well-written and clearly structured, and will likely serve as a valuable resource for interested researchers and practitioners. In my opinion, the work is suitable for publication after minor revisions.

**2 Discussion by themes**

- Performance: The discussion on performance could be expanded slightly. For instance, the following two references, although focused on CPU implementations, could provide useful context:

  - https://doi.org/10.1177/10943420211006169
  - https://doi.org/10.1016/j.compfluid.2021.104946

  If available, including performance estimates of a code such as FastEddy (https://doi.org/10.1029/2020MS002100) in Figure 4, or at least discussing them in the text, would be valuable. Finally, the issue of memory boundedness and the need for advanced streaming patterns to reduce memory requirements could also be discussed in this section.

  We have added the proposed references and additional discussion on memory boundedness in section 3.8.

- LBM for ABLs: The review effectively highlights the potential of LBM for simulating kilometre- scale domains. However, I would suggest adding brief discussions on the following aspects:

  - Meso-scale simulations: LBM is likely constrained to micro-scale simulations due to its reliance on isotropic meshes, making meso-scale applications inefficient. This point seems to be missing from Section 3.5.

    We have added an remark in section 3.5

  - Coupling with meso-scale models: Related to the previous item, it would be useful to discuss the potential need for coupling LBM with meso-scale models and possible approaches for achieving this.

    We have added some discussion on this approach.

  - The study https://doi.org/10.1063/5.0039516 introduces an anelastic approximation within an LBM framework. This work, and the potential advantages of such an approach, merit at least a brief mention and discussion.

We added some discussion on the proposed paper.

– A genuine question: are there intrinsic limitations within the LBM framework that hinder simulations of non-neutral atmospheres, or the inclusion of humidity and other relevant processes? These might relate to the constant sound-speed assumption, the weakly compressible formulation, or the idealized thermodynamics typically adopted.

The standard LBM is limited to small changes in temperature and density, i.e. the Boussinesq approximation. However, there are many ways to extend the valid range of conditions, for example the approach to use the anelastic approximation by Feng or the free energy model for multiphase flows. Several extensions for fully compressible flows have also been proposed.

- LBM theory:

  – Another question: higher-order stencils (beyond order 27) are not discussed. Could they offer any advantages for atmospheric flow simulations?

  We briefly mention higher order stencils in section 2.3. Due to their instability and very large memory footprint we dont believe that high order stencils offer any advantages over hybrid or DDF approaches.

  – Many Navier–Stokes-based ABL solvers include transport equations for humidity (or other scalars) and for kSGS . Implementing these in LBM, especially alongside a double-distribution- function (DDF) approach, might be prohibitive due to memory constraints. This trade-off could be worth mentioning.

  We added some remarks on the issue in section 2.3.

- Turbulence models in LBM:

  – The focus on LES is well justified for atmospheric flows, but a short discussion on the potential use of RANS-type eddy-viscosity closures in LBM, and their possible efficiency benefits, would strengthen the review.

  We have added a very short mention of the possibility to use RANS-type closures.

  – In Section 2.1.6, it would be helpful to comment on how the choice of subgrid-scale model affects performance. Not all SGS models are fully local, and some rely on additional trans- port equations for $k_{\mathrm{SGS}}$ . In a DDF framework, this would require solving yet another set of populations, significantly increasing memory usage — this limitation deserves mention (see also the related note in the LBM theory subsection).

  We have added some discussion in section in 2.1.6

– In Section 3.2, a dedicated discussion on the use of fourth-order limiters in the cumulant space could be beneficial. It is often argued that these limiters behave as implicit SGS models and therefore should not be used in combination with explicit SGS closures.

We have added some discussion on the topic at the end of section 3.2. However, to the best of our knowledge, this topic has not yet been examined in detail.

**3  Small comments**

- L.82: Avoid using superlatives such as "excellent description."

- L.94: It would be interesting to discuss the feasibility of incorporating more complex equations of state within LBM.

- L.175: Replace "an" with "a."

- L.260: The statement that LES is required for ABL flows may be too strong; RANS models could also be employed.

- L.282: Referring to "turbulence models" here creates some ambiguity. Since the discussion focuses on LES, "subgrid-scale models" would be more appropriate.

We have clarified the above mentioned statements.

- Section 2.1.6: Is there a recommended approach overall?

While we cannot give a general recommendation, we give a bit more detail in the discussion, particularly in terms of computational cost.

- Section 2.2: I may be misunderstanding this part, but in my view, the much smaller time steps in LBM are primarily due to the sound-speed-related restriction, rather than to the choice of explicit versus implicit schemes. This point could be clarified.

We have removed the statement and only point to the more detailed discussion in the theory section.

- L.659: Could you elaborate on the assumption that urban flows reduce the need for wall models? Boundary layers developing along buildings are likely to be severely under-resolved.

We have added some clarifying remarks and a source. Urban flows often feature complex flow separation and other features, making wall models ill-suitable.

- L.739: Typo — "different" should replace "difference."

- L.742: Typo — "date" should be "data."

We have fixed the typos.

- A recent paper addressing wind farms in non-neutral ABLs could be included in the discussion: https://papers.ssrn.com/sol3/papers.cfm?abstract_id=4441495.

  We have included the study in the discussion.